# Contribution of rare and common variants to intellectual disability in a sub-isolate of Northern Finland

Mitja I. Kurki[1,2,3], Elmo Saarentaus [3], Olli Pietiläinen[2,4], Padhraig Gormley [1,2], Dennis Lal[1,2], Sini Kerminen[3], Minna Torniainen-Holm[3,5], Eija Hämäläinen[3], Elisa Rahikkala[6,7,8], Riikka Keski-Filppula[6,7,8], Merja Rauhala[9], Satu Korpi-Heikkilä[9], Jonna Komulainen–Ebrahim[10], Heli Helander[10], Päivi Vieira[10], Minna Männikkö[11,12], Markku Peltonen[5], Aki S. Havulinna[3,5], Veikko Salomaa [5], Matti Pirinen [3], Jaana Suvisaari [5], Jukka S. Moilanen[6,7,8], Jarmo Körkkö[9], Outi Kuismin[3,6,7,8], Mark J. Daly[1,2,3,13,14] & Aarno Palotie[1,2,3,13,14]

The contribution of de novo variants in severe intellectual disability (ID) has been extensively studied whereas the genetics of mild ID has been less characterized. To elucidate the genetics of milder ID we studied 442 ID patients enriched for mild ID (>50%) from a population isolate of Finland. Using exome sequencing, we show that rare damaging variants in known ID genes are observed significantly more often in severe (27%) than in mild ID (13%) patients. We further observe a significant enrichment of functional variants in genes not yet associated with ID (OR: 2.1). We show that a common variant polygenic risk significantly contributes to ID. The heritability explained by polygenic risk score is the highest for educational attainment (EDU) in mild ID (2.2%) but lower for more severe ID (0.6%). Finally, we identify a Finland enriched homozygote variant in the CRADD ID associated gene.

[1] Psychiatric & Neurodevelopmental Genetics Unit, Massachusetts General Hospital, Boston, MA 02114, USA. [2] The Stanley Center for Psychiatric Research, The Broad Institute of MIT and Harvard, Cambridge, MA 02142, USA. [3] Institute for Molecular Medicine Finland (FIMM), University of Helsinki, FI-00014 Helsinki, Finland. [4] Department of Stem Cell and Regenerative Biology, University of Harvard, Cambridge, MA 02138, USA. [5] National Institute for Health and Welfare, 00271 Helsinki, Finland. [6] PEDEGO Research Unit, University of Oulu, FI-90014 Oulu, Finland. [7] Medical Research Center, Oulu University Hospital,, University of Oulu, FI-90014 Oulu, Finland. [8] Department of Clinical Genetics, Oulu University Hospital, 90220 Oulu, Finland. [9] Northern Ostrobothnia Hospital District, Center for Intellectual Disability Care, 90220 Oulu, Finland. [10] Department of Children and Adolescents, Oulu University Hospital, Medical Research Center Oulu, University of Oulu, FI-90029 Oulu, Finland. [11] Center for Life Course Health Research, Faculty of Medicine, University of Oulu, Oulu, Finland. [12] Infrastructure for population studies, Faculty of Medicine, University of Oulu, Oulu, Finland. [13] Analytic and Translational Genetics Unit, Department of Medicine, Massachusetts General Hospital, Boston, MA 02114, USA. [14] Department of Neurology, Massachusetts General Hospital, Boston, MA 02114, USA. These authors jointly supervised this work: Mark J. Daly, Aarno Palotie. Correspondence and requests for materials should be addressed to A.P. (email: aarno.palotie@helsinki.fi)

ntellectual disability (ID) is a relatively common disorder characterized by deficits in both intellectual and adaptive functioning in conceptual, social and practical domains. A diagnosis of ID requires deficits in a broad range of intellectual functions, deficits in adaptive functioning resulting in failure to meet developmental and sociocultural standards for personal independence and social responsibility, and an onset during the developmental period[1]. The population prevalence estimates of ID varies between 1 and 3% and is clearly lower (<0.5%) for more severe forms of ID (IQ < 50) than for mild forms[2].

While genome-wide studies using microarrays and exome sequencing have identified a prominent role of de novo copy number variations (CNVs), INDELs and single nucleotide variants in mostly severe ID with reported diagnostic yields of 13–42%, their role in mild ID is less studied but expected to have a less prominent role[3,4]. Intriguingly siblings of mild ID individuals have low IQ compared to the general population whereas the IQ of siblings of severe ID individuals do not differ from the general population[5]. Reichenberger et al.[5] conclude that mild ID represents a low extreme in a normal distribution of IQ, while severe ID is a distinct condition with different etiology[5].

The observation that intellectual disability has a high co-morbidity with other neurodevelopmental and neuropsychiatric diseases such as autism, schizophrenia, and epilepsy has stimulated the hypothesis that these diseases might, in part, have shared genetic backgrounds and thus alterations in the same pathways[6].

One strategy to shed light on the genetic background of diseases is to use populations where the incidence of the trait is higher, and/or where the population history provides benefits for variant identification. Finland is a well-characterized genetic isolate where the small size of the founder population, subsequent bottleneck effects, and genetic drift have caused an enrichment of some rare and low-frequency variants as compared to other

European populations[7,8]. In a population with a recent bottleneck, such as Finland, variants conferring a high risk for a disease with reduced fecundity can exist at markedly higher frequencies than in older populations because negative selection has not had time to drive down the allele frequencies, and therefore these variants are easier to associate to a disease[9].

Interestingly, ID (Fig. 1) and other neurodevelopmental and neuropsychiatric diseases (NDD) like schizophrenia have a higher prevalence in North-Eastern Finland as compared to South-Western Finland[10,11]. It has been hypothesized that such a pattern is related to the recent bottlenecks of these regions. The Eastern and Northern parts of Finland were inhabited more permanently only after the internal migration of small groups in the 16th century while Southern coastal regions were already more populous (Fig. 1)[12]. The regional genetic differences between the early and late settlements (east-west and north-south) can be clearly recapitulated from genome-wide common SNP data[13–15].

The aforementioned Finnish population history and the observation of geographical differences in the prevalence of neurodevelopmental diseases in Finland motivated us to initiate the Northern Finland Intellectual Disability (NFID) study, a geographically based cohort of ID patients and their family members recruited from specialty clinics in the two most Northern provinces of Finland. The only study exclusion criterion was having a known or suspected genetic or environmental cause for the phenotype and therefore the majority of our patients have the most common mild form of ID. Here we describe a comprehensive genetic characterization of 442 independent NFID patients with unknown disease etiology, enriched for mild (51.4%) forms of ID.

We then examined the genetic architecture of this ID cohort that has undergone a population bottleneck and has a high proportion of mild ID cases. We studied the contribution of rare

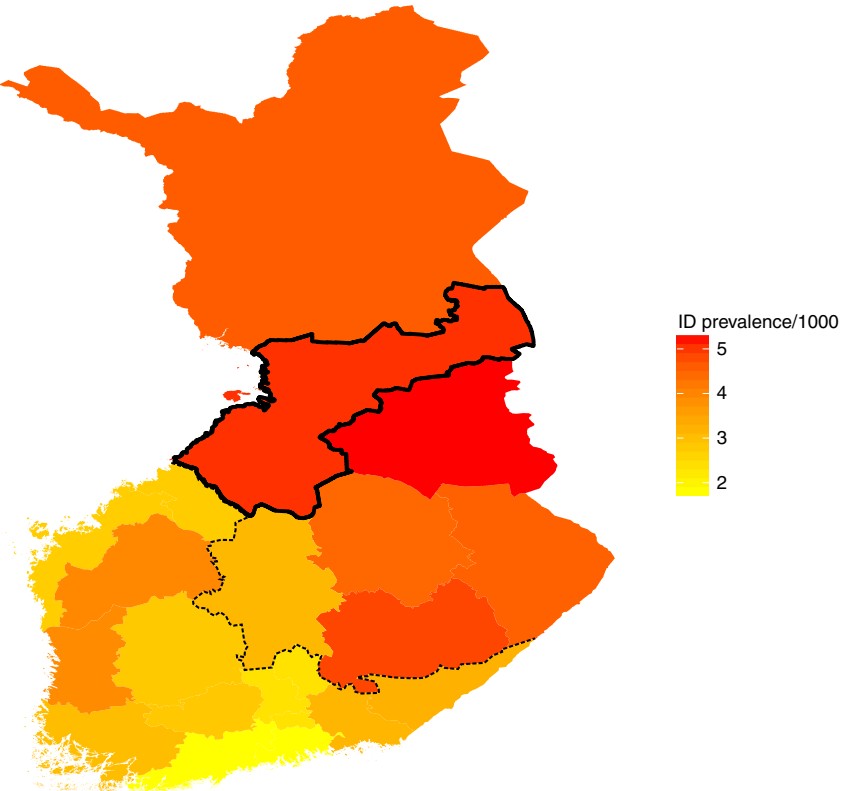

**Fig. 1** ID prevalence estimates in different municipalities in Finland. The primary NFID collection municipalities of Northern Ostrobothnia, Kainuu and Lapland are outlined in solid black. The approximate boundary between early and late settlements is shown with a dashed line

variants using exome sequencing, common variant polygenic risk scores, and CNVs using genome-wide association study (GWAS) arrays, each for different ID severity categories. We then compared the common and rare variants observed in the ID cohort to a large collection of pre-existing Finnish exome ($n = 11,311$) and GWAS array data ($n = 11,699$).

We also analyzed the geographical distribution of the polygenic risk variant load of educational attainment, IQ and schizophrenia across different parts of Finland. Finally, to explore the broader phenotypic impact of identified variant categories and individual variants in the NFID cohort, we compared the identified variants to 640 exome-sequenced individuals with cognitive impairment, schizophrenia (SCZ) or autism spectrum disorder (ASD).

We show that damaging variants in known ID genes are more often identified in more severe ID than mild ID. We further show that polygenic common variant burden is associated with all severity forms of ID and the polygenic risk seems to act in additive manner with rare damaging variants in known ID genes.

## Results

**Regional ID prevalence in Finland.** We first estimated the regional prevalence of ID in Finland using the social security disability benefits register. We observed a higher prevalence of individuals receiving disability benefits for ID in the Eastern and Northern parts of Finland as compared to Southern and Western Finland (Fig. 1). The highest prevalence was observed in Kainuu and North-Ostrobothnia, two of the primary municipalities of the NFID patient collection (Fig. 1).

**Mutations in known genes causing cognitive impairment.** After joint genotype calling and quality control, we analyzed the exomes of 442 independent ID patients (Tables 1 and 2) and

2206 genetically matched population controls. Out of the 442 independent patients we had exome data for 138 full trios, 133 duos and the remaining 171 patients were cases only. To identify individuals with a potential causative variant in the exome analysis, we first searched for damaging missense or protein truncating variants (PTV) in 818 known developmental delay genes (see Materials and Methods and gene list in Supplementary Data 1). For genes where autosomal recessive inheritance has been reported, only homozygote variants were considered. Within these 818 genes we identified a Likely pathogenic mutation in 64 patients (Supplementary Data 2, see Supplementary Data 3 for clinical details of each patient). For the subset of individuals for which we had parental exome available (138 trios and 133 duos), we further filtered the list of Likely pathogenic variants by not inherited from a parent without learning disability. This step filtered 5/24 likely pathogenic variants in trios and 0/15 in duos (Supplementary Data 2). We also excluded 1 Likely diagnostic variants when the clinical phenotype was clearly different (assessed by clinical geneticist) than reported in the literature. After these filterings we identified Likely pathogenic diagnosis for 59/422 patients in exome sequencing.

When comparing the rate of likely pathogenic variants to the 2206 genetically matched controls, we observed the strongest enrichment in PTV variants (OR: 10.94, 95% CI: 4.89–26.21, $p$: 2.7e−10, Fisher's exact test) followed by dominant acting (OR: 5.47, 95% CI: 3.19–9.38, $p$: 7.5e−12, Fisher's exact test) and recessive (OR: 1.83, 95% CI: 0.7–4.30, $p$: 1.4e−1, Fisher's exact test) constrained/damaging missense variant classes (Fig. 2).

**Variants in novel cognitive impairment genes.** Given that ~86% of our cases did not have a variant affecting a known NDD gene, we wanted to assess if there was a burden of rare variants outside of known genes. We performed an enrichment analysis of variants that were either PTV or constrained damaging missense (MPC > 2) variants and not observed in the non-Finnish GnomAD samples or in our internal Finnish controls (different controls used for filtering and enrichment, see Table 2). First, we verified that there was no spurious enrichment of variants caused by stratification or batch effects by analyzing if there was an enrichment of synonymous variants (not observed in GnomAD or our Finnish controls) between cases and controls. No such enrichment was observed, suggesting that QC and case control matching were successful (Fig. 3).

Dominant PTVs in high pLI genes (OR: 2.65, 95% CI: 2.02–3.47, $p$: 2.2e−12, Fisher's exact test) and constrained damaging missense variants not seen in GnomAD or Finnish controls within novel genes (OR: 1.64, 95% CI: 1.29–2.08,

### Table 1 Patient clinical characteristics in the NFID sample

| ID severity | Mild | Moderate | Profound | Undefined |
|---|---|---|---|---|
| Total | 259 | 126 | 72 | 41 |
| Female % | 37% | 41% | 36% | 44% |
| Epilepsy | 28 (11%) | 19 (15%) | 37 (51%) | 9 (22%) |
| ASD | 22 (8%) | 17 (13%) | 25 (35%) | 3 (7%) |
| Behavioral impairment | 38 (15%) | 28 (22%) | 15 (21%) | 0 (0%) |
| Psychotic disorder | 24 (9%) | 16 (13%) | 8 (11%) | 0 (0%) |
| Dysmorphism | 85 (33%) | 62 (49%) | 41 (57%) | 13 (32%) |

*ASD* autism spectrum disorder

### Table 2 Cohorts used in the analyses

| Analysis cohort | Analyses | Cases | | | | | Controls |
|---|---|---|---|---|---|---|---|
| | | N | ID | ASD | SCZ | EPI | N |
| NFID (primary cohort) | EXOME | 442 | 442 | 62 | 47 | 92 | 2206[b] |
| | PRS | 439 | 439 | 62 | 47 | 86 | 2195[b] (matched) 14,816 (total) |
| | CNV | 433 | 433 | 57 | 47 | 84 | 1100[b] |
| NFNDD (Northern Finland NeuroDevelopmental Disorder) | EXOME association analysis | 314 | 17 | 40[a] | 239 | 26 | 1548 |
| SFNDD (Southern Finland NeuroDevelopmental Disorder) | EXOME association analysis | 322 | 14 | 73[a] | 211 | 33 | 1594 |
| SISU controls | EXOME variant filtering | — | — | — | — | — | 5922 |

Sample sizes are the numbers used in the analysis after quality control. The number of individuals in the analyses are after QC and after related individuals have been removed
*ASD* autism spectrum disorder, *SCZ* schizophrenia, *EPI* epilepsy, *PRS* Polygenic Risk Score, *CNV* copy number variant
[a]Comorbidities were not available from ASD cohorts
[b]None of the controls from the FINRISK study have any NDD (intellectual disability, autism, schizophrenia) or epilepsy and all controls are genetically matched to cases

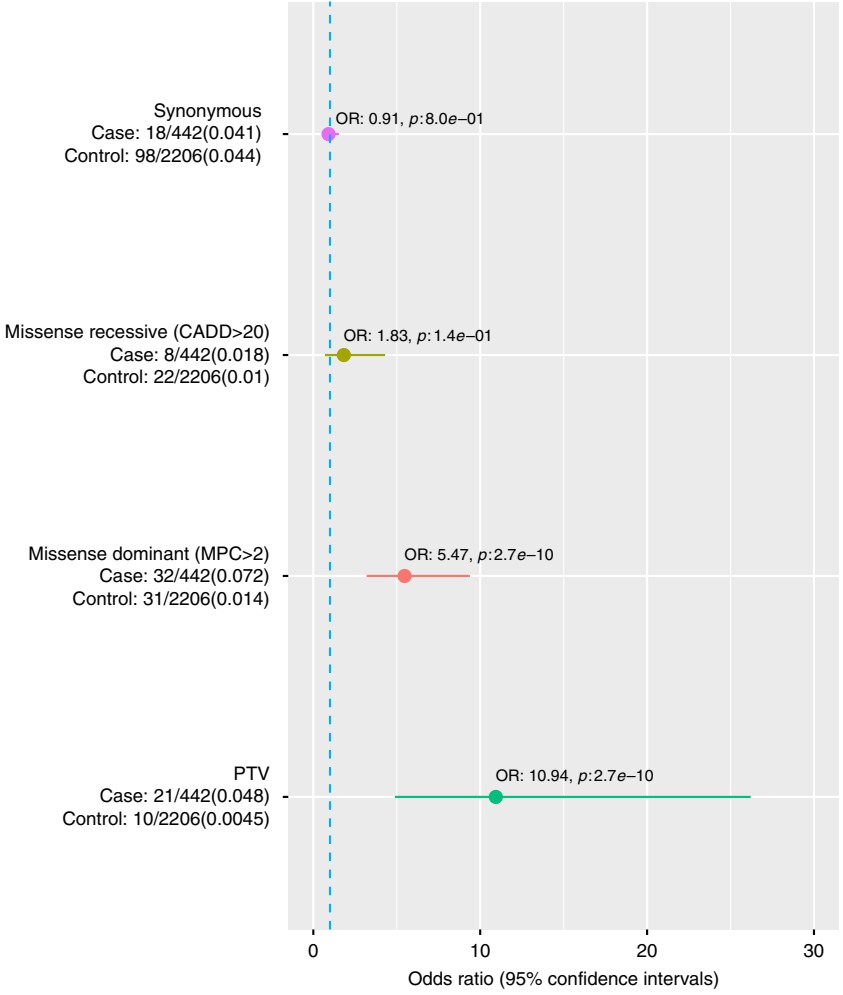

**Fig. 2** Enrichment of likely pathogenic and synonymous variants in known ID genes. Enrichment of variants in 818 known ID genes (Methods) in NFID cases compared to genetically matched controls. Heterozygotes were counted only for those genes for which a dominant inheritance mode is reported. The number of carriers and total individuals are given on the left and in parenthesis the proportion of carriers. Circles indicate the odds ratio (OR) and lines indicate 95% confidence interval of the OR. The synonymous variant identification comparison was performed to assess if possible differences in the variant identification rate due to batch/capture differences were adequately controlled. PTV protein truncating variant, CADD Combined Annotation-Dependent Depletion pathogenicity score, pLI probability of loss of function intolerance. Source data are provided as a Source Data file

$p$: 5.9e−5, Fisher's exact test) were significantly enriched in cases (Fig. 3). The signal for PTV variants was almost exclusively in genes intolerant of PTV-mutations (pLI < 0.95, OR: 1.16, 95% CI: 0.94–1.44, 1.7e−1; Fig. 3). Homozygous PTVs (likely complete knockout of a gene) in novel genes were over twofold enriched in cases, but were not statistically significant (OR 2.51, 95% CI: 0.40–11.78, $p$: 1.8e−1, Fisher's exact test; Fig. 3).

**Copy number variants.** After QC (see Methods), we assessed the contribution of likely diagnostic CNVs in 433 NFID patients and 1100 genetically matched controls. Deletions of any type (>100 kb) were observed slightly more often in cases than in controls (OR: 1.3 (CI 0.94–1.7) $p$: 0.098, Fig. 4a: $n_{cases} = 85$ (19.6%), $n_{controls} = 177$ (16.1%)). However, large deletions (>500 kb) were more frequent in cases, regardless of chromosomal location (OR 4.4 (CI 2.4–8.3) $p$: 4.8e−7 (Fisher's exact test), NFID: 31 individuals (7.2%), controls: 19 individuals (1.8%); Fig. 4b). CNVs that have previously been associated with syndromes, or deleted a known ID gene, were strongly enriched in cases (OR: 26.5 (CI 6.4–233.9), $p$: 4.4e−10, Fisher's exact test) 20 patients (4.6%) vs. two population controls (0.02%); Fig. 4a).

Using our classification algorithm, we identified a Likely pathogenic CNV in a total of 29 cases (Supplementary Data 4). Large deletions (>1 Mb) were the most commonly identified likely pathogenic CNVs (21 cases, 4.9%; 7 controls 0.6%) (Fig. 4b). A total of 17 cases (3.9%) and two controls (0.2%) carried a CNV overlapping a region previously linked to syndromic ID (Fig. 4a). A single 4.7 Mb duplication meeting pathogenicity criteria was detected, overlapping the well-established Prader-Willi/Angelman syndrome region at 15q11-q13. The syndromic CNVs identified in controls were non-ID associated (12p13.33 deletion) and a region with known variable phenotype (22q11 duplication syndrome[16]). Known ID-associated gene was deleted in 14 cases (3.2%) and 0 controls (Fig. 4a). The distribution of duplication sizes is presented in Fig. 4c.

As a pathogenic CNV was observed in only 29 cases (6.7%), we analyzed if there was an excess of smaller deletions in genes intolerant of PTV variations not previously associated with cognitive phenotype. After removing likely pathogenic CNV types, we observed such deletions in 12 cases (2.8%) and 9 controls (0.8%) (OR (CI 1.3 - 9.3), $p$: 5.8e-3, Fisher's exact test) (Fig. 4a and Supplementary Figure 5).

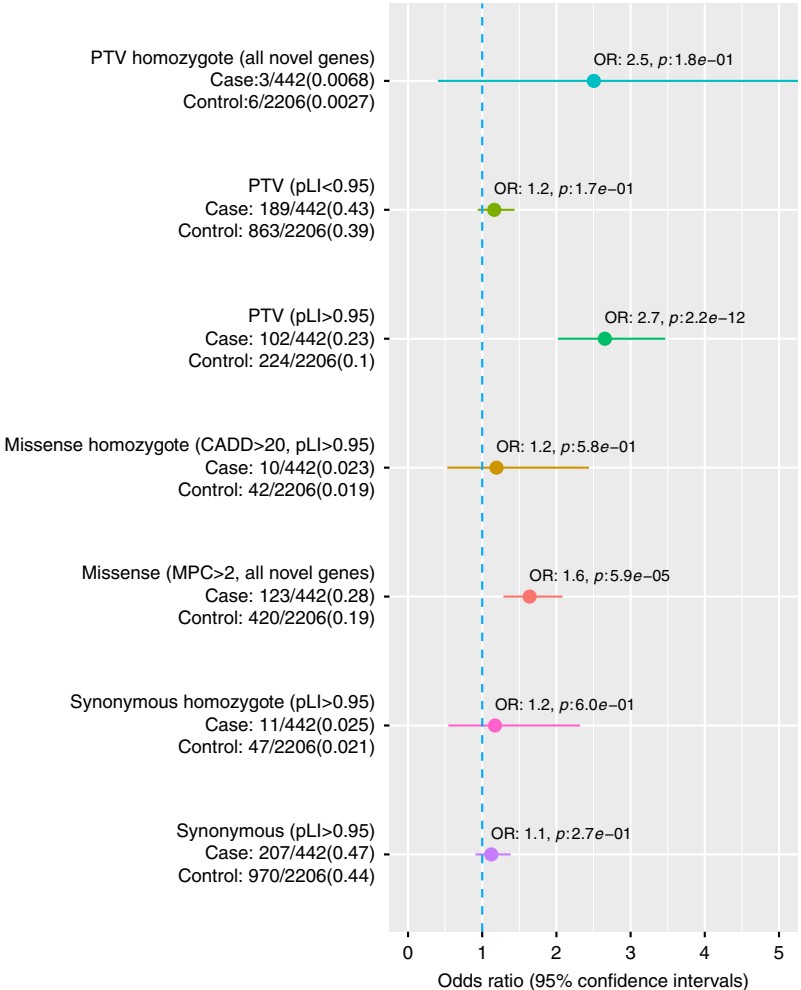

**Fig. 3** Enrichment of rare variants in genes not previously associated with NDDs. Variants not observed in GnomAD or Finnish controls in novel ID genes in cases compared to genetically matched controls (see Methods). Rate in each variant category is first estimated for all novel genes and then after subsetting to only novel high pLI genes. All missense variants are predicted to be deleterious (MPC > 2, see Methods). On the left the number of carriers and total individuals are given and in parenthesis the proportion of carriers. Circles indicate the odds ratio (OR) and lines indicate the 95% confidence interval of the OR. The synonymous variant identification comparison was performed to assess if possible differences in the variant identification rate due to batches/capture differences were adequately controlled. PTV protein truncating variant, CADD Combined Annotation-Dependent Depletion pathogenicity score, pLI probability of loss of function intolerance. Source data are provided as a Source Data file

**Total genetic diagnosis rate**. After combining exome and CNV data, we identified a likely diagnosis for 80 (18.5%) patients (Fig. 5a) of the 433 patients with both exome and CNV data available. The strongest risk factor was having a PTV in a known neurodevelopmental disorder gene (OR 11.1, 95% CI: 4.1–38.1, p: 3.0e−8, Fisher's exact test) followed by a likely pathogenic deletion (OR 8.7, 95% CI: 4.0–21.1, p: 6.5e−10, Fisher's exact test) and a constrained missense variant in a known developmental disorder gene (OR 5.8, 95% CI: 3.0–11.6, p: 9.5e−9, Fisher's exact test) (Fig. 5b). We then analyzed if there was a signal from damaging variants (PTVs, missenses MPC > 2 or CNVs) outside of known ID-associated genes (termed Other high impact variants). We observed a significant enrichment of Other high impact variants in cases vs. controls (Fig. 5a). PTVs (OR 3.0, 95% CI: 2.2–4.2, p: 1.7e−12, Fisher's exact test) and deletions (OR 3.5, 95% CI: 1.3–9.4, p: 5.8e−3, Fisher's exact test) in high pLI genes as well as constrained missense variants (OR 1.7, 95% CI: 1.3–2.2, p: 6.7e−5, Fisher's exact test) were significantly enriched in the Other high impact variants category (Supplementary Figure 6A).

As much less is known about the genetic architecture of mild ID as compared to the more severe ID diagnoses[3], we assessed if rare variants in the same known genes contribute equally to mild and severe forms of ID. For the analysis we combined the moderate, severe and profound ID patients in a severe category. The overall rates of Likely pathogenic (OR 5.4, p: 1.8e−9, Fisher's exact test) and Other high impact variants (OR 2.3, p: 6.5e−7, Fisher's exact test) were significantly higher in the mild group than in controls (Supplementary Figure 7A). However, severe IDs had significantly higher (OR: 2.4, p: 7.0e−4, Fisher's exact test) proportion of Likely pathogenic variants in known ID genes as compared to mild ID (Fig. 5c, Supplementary Figure 8). For CNVs, the diagnostic rate did not follow the same pattern of increased likely pathogenic CNV in more severe cases than in mild cases (Fig. 5d). This is likely because a large fraction of ID patients who had a chromosomal abnormality had been identified in previous clinical cytogenetic analyses and excluded from this study.

As dysmorphic features are present more often in severe ID than in mild ID, we analyzed if Likely pathogenic variants would be found more often in more severe ID due to dysmorphisms and not due to more severe ID. We repeated the enrichment analysis of our variant classification while restricting only to patients with

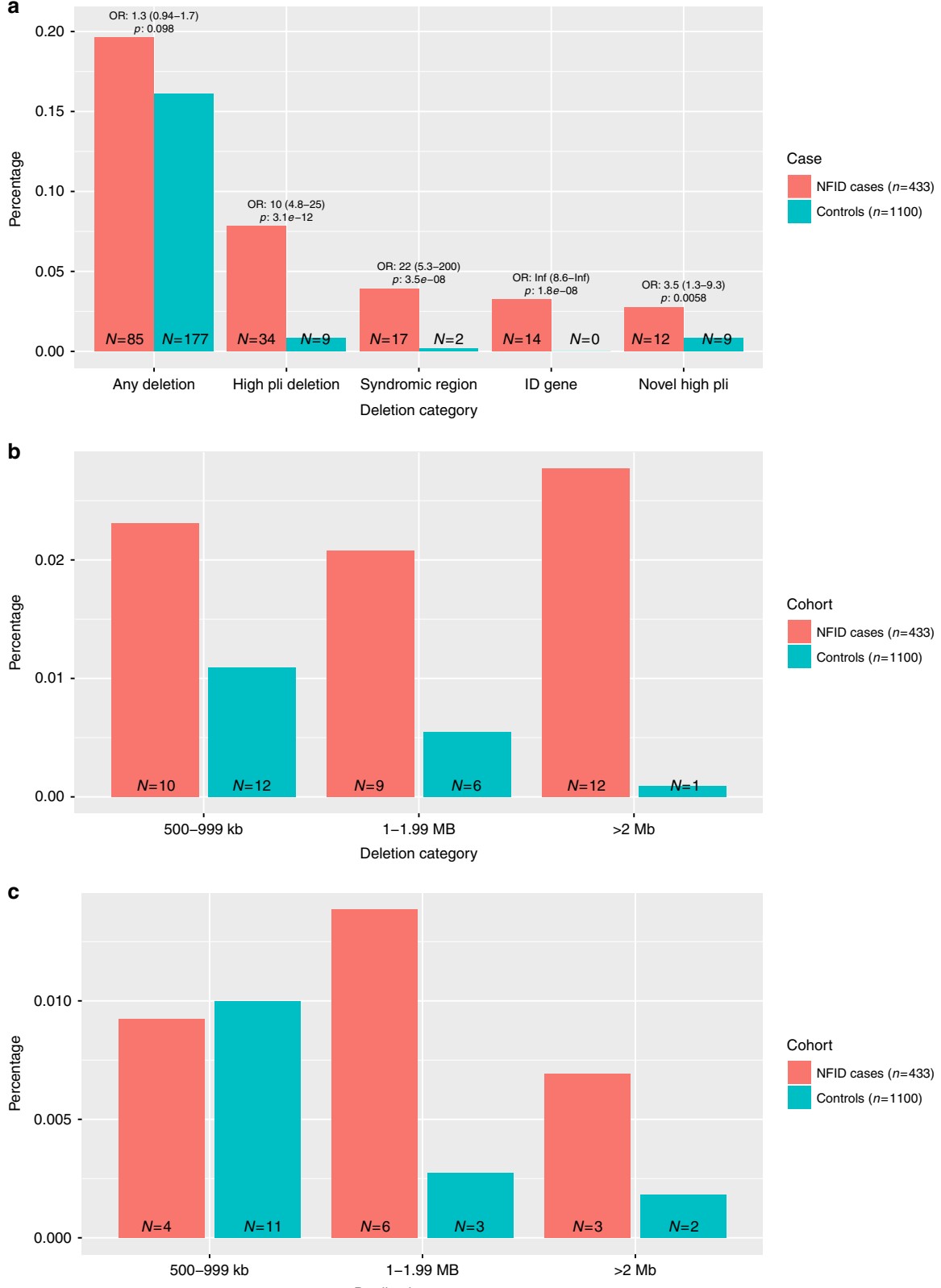

**Fig. 4** Distribution of different deletion categories. **a** Deletion categories in ID patients showed enrichment for deletions (>100 kb) in general, and specifically in deletions covering syndrome regions (as defined by the DECIPHER database), deletions that are located in an ID-associated gene region (see above) or CNVs deleting a gene intolerant of protein truncating mutations (pLI > 0.95). **b** Size distribution of Deletions by size bins, showing enrichment at all sizes > 500 kb . **c** Size distribution of Duplications by size bins > 500 kb, showing an enrichment for sizes > 1 Mb. Source data are provided as a Source Data file

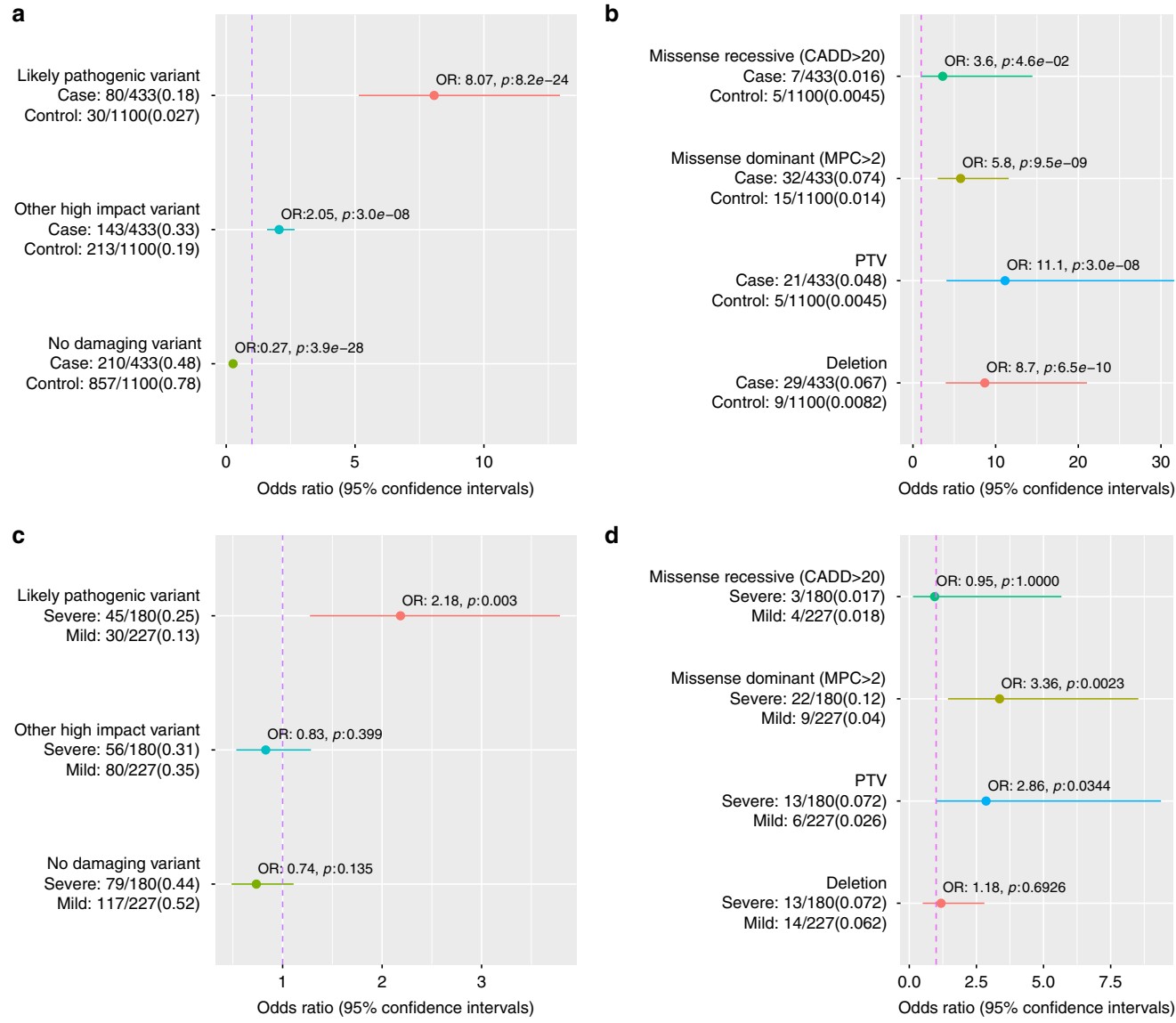

**Fig. 5** Comparison of the total rate of different classes of variants. Comparison was performed in cases vs. genetically matched controls and in mild vs. more severe ID individuals for which both exome and CNV data was available. On the left the number of carriers and total individuals are given and in parenthesis the proportion of carriers. Circles indicate odds ratio and lines indicate 95% confidence intervals of the odds ratio estimate. **a** Total genetic diagnostic rate. **b** Variant classes in "Likely pathogenic" variant categories. **c** Comparison of the rate of identifying different classes of variants in mild vs. severe (moderate and profound ID combined) patients. **d** Comparison of the rate of variant types in "Likely pathogenic" category between mild and more severe forms of ID (moderate, severe and profound ID combined). Constrained missense (MPC > 2) variants were analyzed in all genes instead of only high pLI genes in C and D as MPC score incorporates regional missense constraint. Source data are provided as a Source Data file

no dysmorphisms ($n = 234$) (Supplementary Figure 9). We observed Likely pathogenic variants in 16/83 (19%) and 17/151 (11%) among severe ID and mild ID patients respectively (OR: 1.88, $p$: 0.12, Fisher's exact test). The rate of identifying Likely pathogenic variants was lower in non-dysmorphic, severe (19%) and mild (11%) ID patients than in dysmorphic severe (34%) and mild (18%) patients. There seems to be a higher rate of Likely pathogenic variants in severe ID patients than mild ID patients even among patients for which no dysmorphic features were recorded, although the difference is less pronounced.

As we did not have parental exome sequencing data on all patients, we wanted to assess if the uncertainty in Likely diagnostic classification affects the result that mild ID would be less affected by de novo/ultra-rare variants in known ID genes. To this end we subset the cases to (1) full trios with confirmed de novo in dominant acting gene (2) duos where we checked that the other parent did not have the variant and (3) all patients with Likely pathogenic variants. The OR in SEVERE vs. mild ID patients having Likely pathogenic variant was OR 2.2 (0.6–9.5), 2.8 (0.9–8.8), and 2.2 (1.3–3.8) in confirmed de novos, duos and all patients respectively. This suggests that some misclassification would not change the conclusion that severe ID patients are more often affected by de novos/very rare variants in known ID genes than mild ID.

**Polygenic common variant load**. As we identified likely causative variant in only 18.5% of the cases, we wanted to study the contribution of the polygenic load of common variants associated to intelligence quotient (IQ), educational attainment (EDU) and

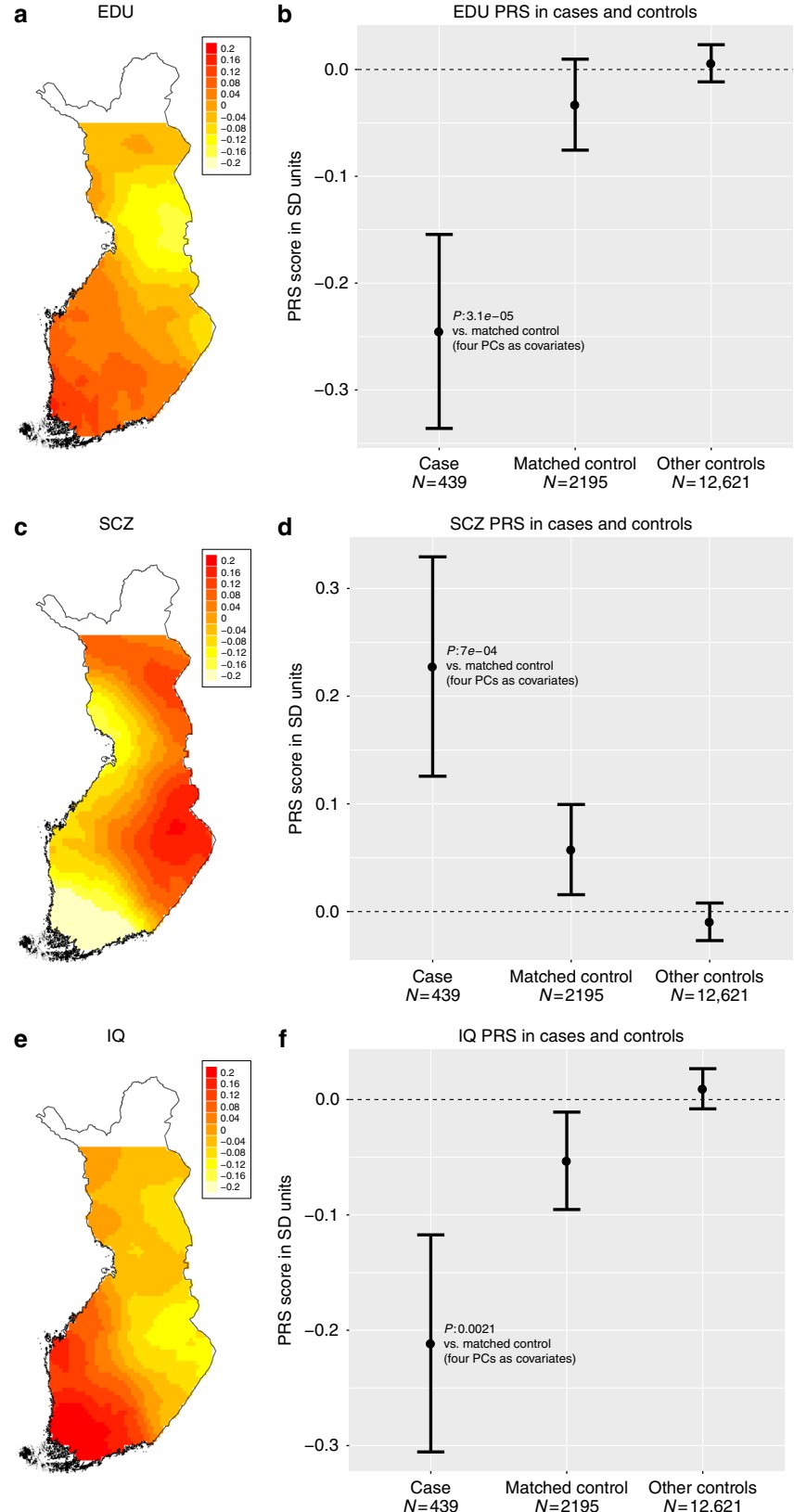

schizophrenia (SCZ) to Northern Finnish ID. There is a partial common variant genetic overlap between cognitive function and schizophrenia[17–19] and therefore we also studied SCZ PRS. We estimated the regional prevalence of SCZ as we did for ID and observed a similar regional enrichment in Northern and Eastern Finland (Supplementary Figure 1).

First, to create a reference for NFID cases, we analyzed whether the geographical distribution of PRSs correspond to the population history of Finland. We genotyped and imputed 14,833 individuals from the population-based FINRISK collection and used all loci with a lead variant *p*-value ≤0.05 in the meta-analyses for schizophrenia, IQ and educational attainment. To

**Fig. 6** Regional distributions of PRSs within Finland and between cases and contols. **a** Locally weighted educational attainment PRS distribution in Finnish population controls whose parents birthplace is within 100 km of each other. **b** Educational attainment PRS in cases and all population controls and genetically matched population controls. **c** Locally weighted schizophrenia PRS distribution in Finnish population controls whose parents birthplace is within 100 km of each other. **d** Schizophrenia PRS in cases and all population controls and genetically matched population controls. **e** Locally weighted IQ PRS distribution in Finnish population controls whose parents birthplace is within 100 km of each other. **f** IQ PRS in cases and all population controls and genetically matched population controls. Error bars indicate 95% confidence intervals around mean. Y-axis is in SD units of PRS standardized to all population controls. Source data are provided as a Source Data file

visualize the geographical distribution of PRSs, we used a distance-weighted polygenic risk score in 2186 Finnish individuals who did not have any neurodevelopmental disorders and whose parents were born within 100 km of each other (see Methods). The PRSs for educational attainment and IQ were lower, and for SCZ higher in the Eastern and Northern part of Finland than in the Southern and Western Finland (Fig. 6a, c, e). Next we asked if the PRSs were associated with ID. All PRSs were significantly associated with the ID phenotype as compared to genetically matched Finnish controls (Fig. 6a, c, e). The PRS for EDU, SCZ and IQ explained 0.94%, 0.55%, and 0.48% of the heritability on the liability scale, respectively (see Supplementary Figure 2 for heritability estimation using varying significance thresholds for locus inclusion).

We next analyzed whether PRS values were different in the different ID groups: mild, moderate, and severe/profound combined. The PRS for EDU was lower and for SCZ higher in the mild ID cases compared to more severe forms, but the differences were not statistically significant (Supplementary Figure 3). Unexpectedly, the IQ PRS in the mild ID group was not significantly different from matched population controls but the most severe ID was different. (Supplementary Figure 3). We also hypothesized that the EDU and IQ PRSs would be lower and SCZ PRS would be higher in patients for which a likely causative mutation was not identified. Thus, we compared the PRSs between cases in different diagnostic categories but did not observe statistically significant differences between groups (Supplementary Figure 4). Importantly, we also subset Likely diagnosed patients to those for which we had confirmed de novo in a known ID gene or diagnostic CNV ($n = 13$). Those 13 patients were just as affected by IQ and EDU PRSs but not by SCZ PRS (Supplementary Figure 4). Assuming we had sufficient power, this suggests that in addition to high penetrance variants, a more general polygenic component (ID and EDU PRS) also contributes to the genetic background of ID.

**Variants enriched in Finland**. Finally, we asked if some variants enriched in Finland might contribute to the Northern Finnish ID phenotype as variants with reduced reproductive fitness can exist in markedly higher frequency in a population with a recent bottleneck[9]. We hypothesized that some of these variants would be associated with ID in the NFID cohort. To identify these variants, we compared the allele frequencies in Finnish samples to the allele frequency in non-Finnish Europeans in the GnomAD database. PTV and missense variants in the range of 0.1–5% (Supplementary Figure 10) were proportionally more enriched compared to other variants. This is in line with our previous observation in smaller datasets[8].

**Dominant variants enriched in Finland**. We first analyzed low frequency and rare (MAF < 0.1% in GnomAD non-Finnish population maximum) single missense and PTV variants enriched at least two-fold in Finland or absent in GnomAD non-Finns excluding singletons (13,483 variants; 12,628 missense, 855 PTV). We identified 396 variants nominally ($p < 0.05$, Fisher's

exact test) associated with ID (Supplementary Data 4). We then aimed to replicate these associations in NDD cases and genetically matched controls from the Northern and Southern Finland (Table 2). After meta-analyzing all three cohorts, we identified 29 variants associated with a $p$-value < 0.001 (Mantel–Haenzel test) (20 variants were found in cases only across the three cohorts). However, none of variants surpassed a Bonferroni multiple-testing correction for 13,483 tests.

**Recessive variants enriched in Finland**. We next asked if some of the enriched PTV or missense variants with low allele frequency in GnomAD (AF < 0.01) were recessively associated with ID. We excluded singleton homozygotes and variants observed as homozygous in GnomAD. After these filtering steps we performed a recessive analysis for 1408 variants (1379 missense variants and 29 PTVs). Eighteen variants were observed as homozygous more than once in cases across the three cohorts but not in controls (Table 3).

We identified a homozygous missense variant in the *CRADD* gene in three independent ID cases. Additionally, we identified one *CRADD* missense homozygote in the Northern Finland NDD case cohort (RAFT meta $p$: 5.75E−8). The variant is over 50 times more frequent in Finland than in non-Finnish Europeans. The variant is located in the DEATH domain through which *CRADD* interacts with other DEATH domain proteins[20]. Another variant in the *HGF* gene achieved a $p$-value surviving Bonferroni correction ($p$: 1.3e−5, RAFT meta), but it was observed only in two cases. Homozygote variants in *HGF* have been identified in consanguineous families ascertained for non-syndromic deafness[21]. Our cases did not have hearing problems. Among the 18 genes with case-only recessive candidate variants we observed significantly more genes that are intolerant of homozygote PTV variation (pRec > 0.8[22]) than expected by chance. A pREC metric was available for 17 of the 18 of the candidate genes, of which eight were intolerant of homozygote PTV variation. In ExAC 4,508 out of 18,241 genes have pRec > 0.8, and therefore we would expect 4.2 genes by chance (binomial test $p$-value 0.046). This suggests that some of the 18 candidate variants are true risk variants for ID (see Supplementary Data 6 for all 57 nominally significant associations).

**Variance explained by different variant categories**. To put the relative contribution of different classes of genetic variation to a context, we estimated the variance explained by each significant category in the case–control comparisons. We added the three *CRADD* homozygotes to the likely pathogenic category as we clearly demonstrated the variant to be a causative recessive variant (Table 3). For likely diagnostic and other high impact variants, we used genetically matched cases and controls for which both exome and CNV data were available (433 cases and 1100 controls). For the polygenic risk score, we used 439 cases and 2195 genetically matched controls. As we observed geographical differences in PRSs within Finland, we corrected the variance explained estimation using the first four PCs. The PRS's contribution to heritability is lower (IQ 0.48%, 95% CI: 0.067–1.25%;

**Table 3 Homozygous Finnish enriched variants observed ≥2 times across NFID and the Southern and Northern Finnish NDD cases and not observed in any controls as homozygous**

| Variant | Gene | Previous evidence | GnomAD Finnish AF | GnomAD pop. Max AF | NFID AF | RAFT meta p | NFID case homs | NFID RAFT p | Population NDD case homs | Population NDD RAFT p |
|---|---|---|---|---|---|---|---|---|---|---|
| 12:94243956 G:A (mis) | CRADD | AR Lissencephaly, ID[25] | 6.01E−3 | 9.15E−4 | 7.83E−3 | 5.01E−8 [a] | 3 | 1.86E−6 | 1 | 9.30E−3 |
| 7:81374424 G:C (mis) | HGF | AR hearing loss (OMIM) | 1.12E−3 | 5.48E−4 | 2.52E−3 | 1.34E−5 [a] | 1 | 3.20E−3 | 1 | 2.54E−3 |
| 12:15784582 T:C (mis) | EPS8 | AR deafness (OMIM) Cognition defects in mice[51] | 9.03E−3 | 1.30E−3 | 6.89E−3 | 1.28E−4 | 2 | 1.28E−4 | 0 | NA |
| 1:220236134 C:T (mis) | BPNT1 | - | 1.07E−2 | 4.83E−3 | 6.89E−3 | 1.34E−4 | 2 | 1.34E−4 | 0 | NA |
| 7:1520077 T:C (mis) | INTS1 | AR ID[26] | 1.30E−2 | 3.05E−3 | 1.71E−2 | 1.95E−4 | 3 | 1.95E−4 | 0 | NA |
| 2:95753239 A:G (mis) | MRPS5 | - | 9.87E−3 | 4.56E−3 | 8.29E−3 | 2.88E−4 | 2 | 2.88E−4 | 0 | NA |
| 10:123844296 C:A (mis) | TACC2 | - | 1.32E−2 | 1.10E−3 | 1.14E−2 | 1.05E−3 | 2 | 1.05E−3 | 0 | NA |
| 1:155028692 C:T (mis) | ADAM15 | - | 8.69E−3 | 2.37E−3 | 1.18E−2 | 2.11E−3 | 1 | 4.84E−2 | 1 | 3.58E−2 |
| 18:14542688 G:A (mis) | POTEC | - | 1.86E−2 | 6.26E−3 | 1.98E−2 | 2.11E−3 | 2 | 1.01E−2 | 1 | 1.65E−1 |
| 21:19651329 G:C (mis) | TMPRSS15 | Enterokinase deficiency (OMIM) | 1.73E−2 | 6.31E−3 | 1.93E−2 | 2.66E−3 | 1 | 1.42E−1 | 2 | 9.85E−3 |
| 15:60789800 T:C (mis) | RORA | AD ID[52] | 1.02E−2 | 9.14E−4 | 1.59E−2 | 4.03E−3 | 1 | 9.35E−2 | 1 | 3.38E−2 |
| 11:6023849 C:T (mis) | OR56A4 | - | 1.69E−2 | 1.82E−3 | 1.61E−2 | 4.45E−3 | 2 | 4.45E−3 | 0 | NA |
| 11:3681309 G:A (mis) | ART1 | - | 1.64E−2 | 3.68E−3 | 1.53E−2 | 5.67E−3 | 1 | 8.43E−2 | 1 | 5.76E−2 |
| 19:56424477 TC:T (frameshift) | NLRP13 | - | 1.65E−2 | 1.28E−3 | 1.40E−2 | 6.07E−3 | 1 | 7.44E−2 | 1 | 8.54E−2 |
| 8:17612739 G:C (mis) | MTUS1 | - | 1.46E−2 | 2.01E−3 | 1.54E−2 | 6.74E−3 | 1 | 9.22E−2 | 1 | 7.45E−2 |
| 1:183520048 A:T (mis) | SMG7 | NMD-components linked to ID[53] | 2.49E−2 | 1.10E−3 | 2.62E−2 | 3.85E−2 | 1 | 2.85E−1 | 1 | 1.58E−1 |
| X:23410887 C:T (mis) | PTCHD1 | x-linked ID/AUTISM[54] | 2.43E−4 | 2.51E−5 | 2.27E−4 | NA | 1[b] | NA | 1[b] | NA[b] |

AR autosomal recessive, AD autosomal dominant, ID intellectual disability, AF allele frequency, mis missense
[a]Significant after multiple testing correction
[b]RAFT statistic not valid for X-chromosome. Both carriers are hemizygote males

SCZ 0.55%, 95% CI: 0.078–1.5%; EDU 0.94%, 95% CI: 0.31–1.92%) than that of pathogenic variants in known genes (4.15%, 95% CI: 2.78–5.77%) or other high impact variants (2.25%, 95% CI: 1.30–3.52%) (Fig. 7a). When comparing the different ID severities, the heritability explained for PRSs was the highest in mild ID for EDU (2.1%, 95% CI: 0.46–4.5%) and smaller in more severe ID (0.52%, 95% CI: 0.04–1.60%). The heritability estimation for all PRSs and ID categories is presented in Supplementary Figure 11.

The variance explained by Likely pathogenic variants in known genes was slightly higher in more severe ID (6.2%, 95% CI: 3.8–9.1%) than in mild ID (4.0%, 95% CI: 1.8–7.1%). This is expected as we observed a significantly lower proportion of Likely pathogenic variants in mild ID (13%) vs. more severe ID (25%) (Figs. 5 and 7b).

## Discussion

Here we have described a comprehensive genetic analysis of an ID cohort from a population with a relatively high prevalence of ID. We studied the contribution of SNVs and INDELs, CNVs, and of a genome-wide common variant polygenic load. Unlike most published studies our ID cohort consists mostly of relatively mild ID cases. We identified a likely pathogenic variant in genes known to be associated with ID in 18% of the cases for which both exome and CNV data were available (Fig. 5a), explaining an estimated 4.2% of the heritability (Fig. 7). Additionally, we observed a significant ~2-fold enrichment of damaging variants/ CNVs in loss-of-function intolerant genes not yet linked to ID, which explained an additional 2.3% of the heritability (Figs. 5a and 7). We then demonstrated that a common variant polygenic load is associated with ID. We observed educational attainment, IQ and schizophrenia polygenic risk scores to be associated with ID explaining an estimated 0.94%, 0.48%, and 0.55% of the heritability, respectively.

We then focused on characterizing the genetic architecture of mild vs. more severe forms of ID and observed that a likely causative variant in known ID genes was significantly more often identified in more severe ID cases than in mild ID cases (Fig. 5c). This suggests that either mild ID has a more complex etiology or that variants in genes predisposing to mild ID are partly different than those predisposing to more severe forms of ID. Our observation is in agreement with epidemiological studies where mild ID has been suggested to represent a highly heritable low end of a normal distribution of IQ whereas severe ID is a distinct condition with different etiology[5]. Therefore, mild ID should have less contribution from de novo and extremely rare variants, which have been the major focus of most genetic studies of ID.

To study the possibly more complex etiology of mild ID, we first showed that the polygenic risk score of low educational attainment, low IQ, and schizophrenia were all higher in the Eastern and Northern parts of Finland, coinciding with the more recent bottleneck and higher prevalence of intellectual disability

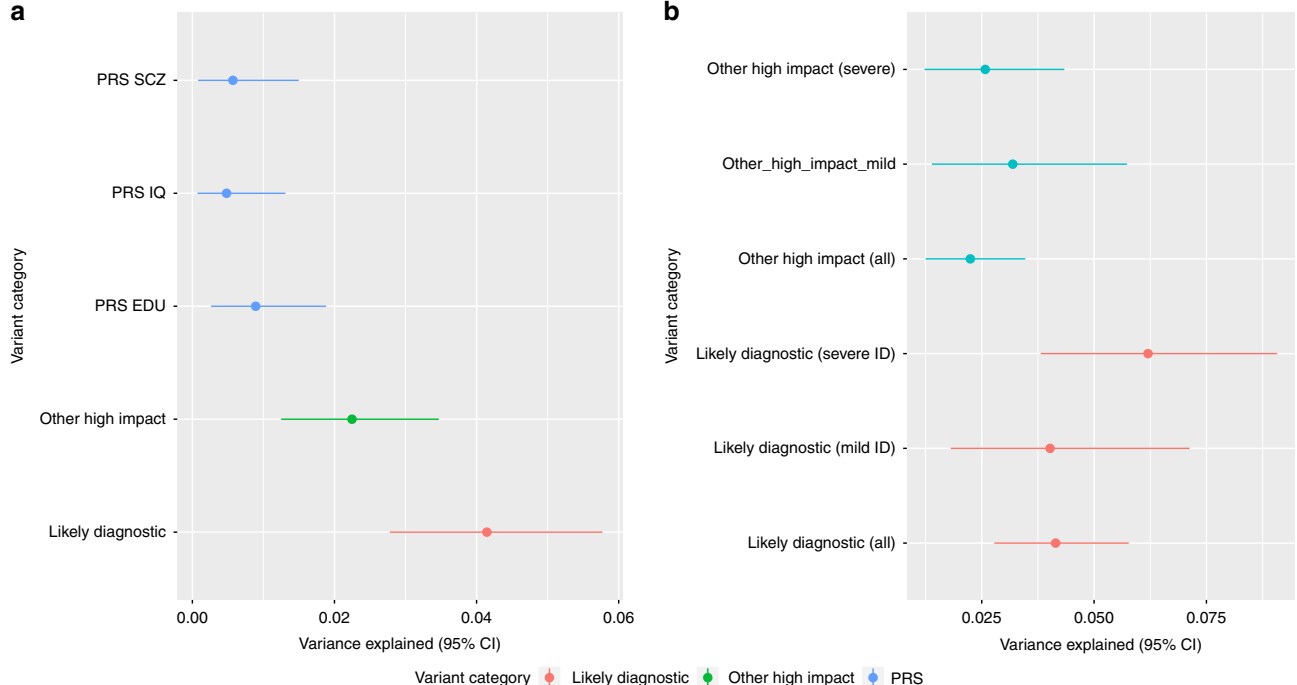

**Fig. 7** Estimate of heritability explained by different variant categories on liability scale. **a** Variance explained by genetic categories in all ID cases. **b** Variance explained delineated by ID severity. Variance explained was estimated by Nagelkerke r2 while controlling for the first four PCs. 95% Confidence intervals of variance explained were estimated by 5000 bootstrap samples. Source data are provided as a Source Data file

and schizophrenia within those regions in Finland[10]. We then showed that the PRS for educational attainment, intelligence and SCZ all were significantly associated with ID in our cohort when compared to the genetically matched control population, thereby demonstrating the contribution of common low-risk variants to intellectual disability. This observation could be in part because most of our ID patients had mild ID. Indeed, the highest heritability explained (2.1%) was observed with EDU PRS in mild ID. The EDU PRS has been reported to explain 2.9% of the heritability of educational attainment in a population sample independent of the original GWAS[23]. Our results suggest that mild ID might be just a continuum of the population distribution of cognitive capacity and support the hypothesis of the polygenic background. The observation that the heritability explained by EDU PRS is clearly smaller in severe ID (0.5%) supports the earlier epidemiological findings that the genetic background of severe ID is different from mild ID[5], where penetrant mutations contribute more to the phenotype.

The PRS for IQ was only slightly below the matched controls in mild ID. This was unexpected. The reason remains speculative, but could be contributed by the fact that the IQ PRS was generated from a smaller study samples ($n = 78,308$) than the EDU score ($n = 293,723$).

After observing a significant association between the common variant load and ID, we hypothesized that PRSs would be different in those individuals in whom a likely pathogenic variant was identified and those where such variants were not identified. However, such a difference was not observed, not even a suggestive trend (Supplementary Figure 4). This observation could be explained by assuming that rare high-risk variants and the common variant load act additively to increase the risk of ID. Another explanation could be that there still might be other unidentified strong or moderate variants explaining the phenotype in many of the cases in which we did not identify a causative variant. We explored this hypothesis by grouping patients into

the Other high impact variant category if they carried a PTV, CNV or damaging missense mutation in loss of function intolerant genes not previously linked to NDDs, but did not observe a difference in PRSs in that group either (Supplementary Figure 4). An additive effect of high impact rare variants and common variant polygenic load has recently been suggested in the genetic etiology of ASD[24], our data suggest a similar genetic architecture for ID.

Finally, we studied if some variants enriched in Finland in the relatively recent bottleneck would be associated with ID in our cohort. We conclusively identified a recessive variant in the *CRADD* gene enriched in Finland in three NFID patients and one NDD patient from the population NDD cohorts (Table 3). The allele frequency of this variant is 50× higher in the Finnish population than in non-Finnish Europeans. Recently recessive variants in *CRADD* have been reported in six patients from four families with megalencephaly, frontal predominant pachygyria, intellectual disability, and seizures[25]. All three of our patients had pachygyria, consistent with previously reported cases[25]. One of the patients identified in Di Donate et al. had Finnish origins and carries exactly the same homozygotic variant as our patients, clearly demonstrating that the variant is a causal for a specific syndrome. We also observed three cases that had the same missense variant in homozygous state in the *INTS1* gene (Table 3). Recently a loss-of-function variants in *INTS1* have been identified in three unrelated moderate to severe ID patients[26]. One of our patients had mild ID and the two others had moderate/severe ID.

In the dominant association analysis of Finnish enriched variants, none of the variants surpassed multiple testing correction (Supplementary Data 5). However, one variant among the top 10 variants, a missense variant in the *DENR* gene, was totally absent in non-Finnish GnomAD individuals, is very rare in the Finnish population but enriched in Northern Finland ($6.3 \times 10^{-4}$ in GnomAD Finns; $9.7 \times 10^{-4}$ in our Northern Controls and $3.1 \times 10^{-4}$ in Southern controls). The variant replicated in the Northern

NDD cohort and was extremely rare in Southern Finnish NDD cases and controls (1/322 in cases and 1/1594 in controls) but had a high OR estimate consistent with associations in NFID and Northern NDD samples. Two *DENR* de novo missense variants have previously been identified in patients ascertained for autism spectrum disorder[27,28]. The case in Neale et al. had an IQ of 67 and the case in Haas et al, had a language delay and poor comprehension. Two of the three *DENR* variant carriers in the NFID cohort had a suspected or confirmed ASD diagnosis. Eight individuals in the population NDD cases were schizophrenia patients. The SCZ cases had low scores on processing speed and verbal learning cognitive tests as compared to population controls (Supplementary Figure 10). ID or autism were not systematically diagnosed in the collection. Further studies are needed to conclusively determine if some of the other identified candidate genes are truly ID associated.

Limitations of the study includes the fact that we used exome sequencing although non-coding rare variants also contribute to the etiology of ID[29]. Also, we did not have exome sequencing for parents for 2/3 of the patients, but we performed sensitivity analyzes on the subset of patients, where we had full trios. These supported the conclusions that common variant polygenic load and rare variants might act additively and that mild ID is less affected by de novo/extremely rare variants in known ID genes (see total genetic diagnosis rate chapter and Supplementary Figure 4).

In conclusion, we demonstrate that a common variant polygenic load is a contributing factor in ID and more broadly characterized the genetic architecture of mild ID, which so far has been understudied. We also show that some damaging variants enriched in frequency in Finland contribute to intellectual disability and provide, yet another example of the power of utilizing population isolates such as Finland in disease gene mapping.

## Methods

**Samples**. Since January 2013 subjects for the NFID (Northern Finland Intellectual Disability) project have been recruited from the Northern Ostrobothnia Hospital District Center for Intellectual Disability Care and from the Department of Clinical Genetics of Oulu University Hospital. In January 2016 the recruitment was expanded to include all pediatric neurology units and centers for intellectual disability care in the special responsibility area of Oulu University Hospital. Subjects of all ages with either intellectual disability or pervasive and specific developmental disorders (ICD-10 codes F70-79 and F80-89, respectively) of unknown etiology were included. Individuals with copy number variations of unknown clinical significance or highly variable phenotypes were also included in order to uncover other possible factors of genetic etiology. Subjects were identified through hospital records and invited via mail to take part in the study. In addition, they were recruited during routine visits to any of the study centers.

The cases have been evaluated and examined clinically by multi-professional teams. Depending on the situation in question the team may consist of psychologist, physician, speech and occupational therapist, physiotherapist, nurse and social worker. Standardized IQ tests that were used included different versions of following tests: Wechsler Preschool And Primary Scale Of Intelligence (WPPSI), Wechsler Intelligence Scale for Children (WISC) and Wechsler Adult Intelligence Scale (WAIS) for adults.

In case of autism spectrum disorder the diagnoses were also based on multiprofessional evaluation and different, clinically used methods such as ADOS (Autism Diagnostic Observation Schedule), ADI-R (Autism Diagnostic Interview), and CARS (Childhood Autism Rating Scale).

All research subjects and/or their legal guardians provided a written informed consent to participate in the study. DNA samples from the participants were extracted primarily from peripheral blood. In a few cases where a blood sample could not be obtained, DNA was extracted from saliva. The ethical committees of the Northern Ostrobothnia Hospital District and the Hospital District of Helsinki and Uusimaa approved the study.

Clinical diagnostic tests varied considerably depending on the subject´s age, clinical diagnosis and phenotype. During the past 20 years, blood and urine metabolic screening tests, chromosome karyotyping, FMR1 CGG repeat analysis, electroencephalography (EEG) and brain computed tomography (CT) or magnetic resonance imaging (MRI) have been routinely performed on almost all individuals with remarkable developmental delay or intellectual disability. Array CGH and whole exome sequencing have been widely used for less than ten and three years, respectively.

**Identification of other neurodevelopmental disorder cases**. We identified individuals with neurodevelopmental disorder (NDD) phenotypes (intellectual disability, schizophrenia, autism and epilepsy; $N = 636$, NFNDD and SFNDD cases in Table 2) among 5904 individuals with exome sequence data in the FINRISK study. FINRISK is a series of population-based health examination surveys carried out every 5 years since 1972 to monitor the risk of chronic diseases[30]. The cohorts have been followed up for disease end-points using annual record linkage with the Finnish National Hospital Discharge Register and the National Causes-of-Death Register.

Additional Finnish NDD cases were included from cohorts of schizophrenia and autism patients sequenced as part of the UK10K-study (i.e. subcohorts UK10K_NK_SCZ, UK10K_KUUSAMO_SCZ and UK10K_ASDFI) and a collection of autism patients from Southern Finland (AUTISM_ASDFI) (see Supplementary Data for cohort descriptions). We genetically matched each NDD case to five exome sequenced controls using the first 2 principal components (PCs). We further divided these cases and controls approximately to Northern Finnish NDD (NFNDD, Northern Finland NeuroDevelopmental Disorder) and Southern Finnish NDD (SFNDD, Southern Finland NeuroDevelopmental Disorder) cohorts based on principal component analysis (PCA).

**Regional prevalence of intellectual disability in Finland**. To estimate regional prevalence of ID and SCZ in Finland, we used The Social Insurance Institution of Finland provides social security coverage for Finnish residents. The Social Insurance Institution of Finland centrally provides all disability pensions in Finland and maintains a database of all residents on a disability pension and the reason for the pension. We requested the number of individuals over 16 years of age receiving a disability pension for ID or schizophrenia (SCZ) at the end of year 2016 in each of the 19 high-level administrative regions in Finland. We divided the number of beneficiaries by the population aged over 16 in each region to get a crude estimate of the relative prevalence of more severe SCZ and ID cases. The prevalence of schizophrenia particularly is higher in more detailed prevalence estimates[11]. Schizophrenia tends to be underdiagnosed in the first years of illness[31], and only 50% of patients with schizophrenia receive a disability pension after 5 years of initial diagnosis[32].

**CNV analysis**. To analyze the copy number variations (CNVs), we performed DNA Chip Array (Illumina HumanCoreExome v 12.0, Illumina PsychArray) based copy number analysis of 497 cases and 504 unaffected family members of the NFID cohort. To assess CNV frequencies in the general population, we used as controls a population-based cohort of 13,390 participants from the FINRISK study[33]. CNV calls in controls were generated using raw data from the Illumina HumanCoreExome v12.0 and v12.1 chips.

CNVs were called using a CNV pipeline powered by PennCNV[34] for sensitive CNV calling. Adjacent CNVs of similar copy number were called as one if the adjoining region between the two calls was ≤20% of the joined CNV. To increase the confidence in the called CNVs, we considered only CNVs supported by at least 10 consecutive probes and which covered a genomic region of at least 100 kb, omitting known CNV artifacts regions[35]. The large regional requirement was set to support analysis across the different DNA chips.

Samples were excluded if they had: (1) a high variance (SD > 0.3) in intensity (1.5% in NFID; 5.6% in FINRISK), (2) a high (>0.005) drift of B allele frequency (0 additional samples in NFID; 0.2% in FINRISK), and (3) CNVs called in excess of 10 for one individual (10 samples in NFID; 8.9% in FINRISK). All called CNVs for the NFID cohort, both for patients and for unaffected family members, were manually curated. For the FINRISK population cohort, CNVs were manually curated if large (>500 kb) or if they fit into a category of interest relevant to study (see Identifying likely pathogenic mutations chapter below). Otherwise, CNVs of controls were rejected if at least 50% of the CNV overlapped a known artifact region[36], or had a poor coverage (≤1.08 SNPs per 10 kb).

**GWAS data processing**. All samples were genotyped in seven batches on either the Illumina CoreExome or Illumina PsychArray, which contains 480,000 common variants. The NFID samples were genotyped in three batches, one with Illumina CoreExome and two with PsychArray. FINRISK population controls were genotyped in five batches using Illumina CoreExome.

We excluded markers that exhibited high missingness rates (>5%), low minor allele frequency (<1%), or failed a test of Hardy–Weinberg equilibrium ($p < 1e-9$). We also excluded individuals with high rates of heterozygosity (>3sd from the mean), or a high proportion of missing genotypes (>5%). To control for population stratification, we merged the genotypes from individuals passing QC with HapMap III data from European (CEU), Asian (CHB + JPT), and African (YRI) populations. We then performed a PCA on this combined data and excluded population outliers not clustering with the Finnish samples

We then merged genotyping batches one-by-one and repeated the QC procedures described above on the merged dataset. To prevent any potential batch effects in the merged data, we also excluded any markers that failed a test of differential missingness ($p < 1e-5$, Fisher's exact test) between the merged batches. Furthermore, during each round of merging we performed a association analysis (using a logistic mixed-model for individuals) between samples from each batch to

identify markers where the minor allele frequency deviated significantly between batches ($p < 1e−5$, score test). Finally, we removed related individuals (identity by descent > 0.185).

We used a custom Finnish imputation reference panel containing 1941 low-pass whole genomes (4.6×) and 1540 high coverage exomes. We used Shape-IT[37] for pre-phasing and Impute-2[38] for imputation.

**Exome sequencing.** NFID cases were exome sequenced at the Broad Institute using Illumina Nextera Rapid Capture Exome-capture kit and sequenced with Illumina HiSeq2000 or 2500. NFID cases were jointly called with a collection of Finnish individuals collected as part of the Sequencing Initiative Suomi (SISU)-study (www.sisuproject.fi). The sequence data processing and variant calling has been described previously[39]. See Supplemental Note 1 for descriptions of cohorts used in the current study.

We filtered samples with estimated contamination > 3% ($n = 590$), chimeric reads > 3% ($n = 51$), samples significantly deviating from other samples within each project/batch on selected metrics (transition/transversion ratio, insertion/deletion ratio, heterozygous/homozygous variant ratio, number of singletons, $n = 243$) and finally included only those with empirically confirmed ≥99% Finnish ancestry (described in Rivas et al.[39]).

We first split the multiallelic variants in to bi-allelic variants. For genotype QC, we set the following genotypes to missing; genotype quality (GQ) < 20, read depth (DP) < 10, heterozygote allelic balance less than 20% or greater than 80%, homozygous reference alt reads ≥10%, alternate allele homozygous reference reads ≥10%. Variants were filtered out if Variant Quality Score Recalibration (VQSR) did not indicate PASS, the $p$-value from a test of Hardy–Weinberg Equilibrium (pHWE) < 1e−9 in controls (in females only in the X chromosome), SNP quality-by-depth (QD) < 2, INDEL QD < 3 or more than 20% of heterozygote calls had allelic balance out of the 20–80% range. To account for the different batches of exome sequencing we required a stringent genotype call rate ≥0.95 in cases and controls separately after genotype QC. All variant and genotype QC was performed using Hail[40] and executed in the Google Cloud dataproc cluster.

Finally, we ensured cases and controls were approximately independent by filtering such that all samples had a pairwise kinship coefficient < 0.0442 to every other sample. We estimated kinship coefficient using King[41] and when possible we always retained cases rather than a related control ($N$ filtered = 1531).

**Variant annotation.** We annotated variants using VEP v.85 and the LOFTEE VEP plugin [https://github.com/konradjk/loftee] to filter likely false positive protein truncating variants (PTV). We considered variant annotations of the canonical (as defined by ENSEMBL) transcript only. A variant was considered to be a protein truncating variant (PTV) if LOFTEE predicted it to be a high confidence loss-of-function variant (stop-gained, splice site disrupting or frameshift) without any warning flags.

**Identifying likely pathogenic mutations.** As a basis for identifying Likely pathogenic variants, we used a gene list curated within the Deciphering Developmental Disorders study (DDD) and a gene list of 93 exome-wide significant genes from the latest DDD study meta-analysis of de novo variants[4]. We downloaded a gene list curated within the DDD study [https://decipher.sanger.ac.uk/ddd#ddgenes] containing 1897 genes with varying degrees of evidence of mutations in those genes causing developmental delays. We further subset the list to only confirmed or probable developmental delay genes contributing to a brain/cognition phenotype. This gene set was further extended by a set of 93 genes with a significant excess of damaging de novo variants in the latest DDD meta-analysis[4]. These two lists resulted in a total of 818 genes (Supplemental Table 1). For each ID patient we searched for PTV or damaging missense (MPC ≥2[42]) variants not observed (as homozygotes in recessive genes) in non-Finnish GnomAD individuals or in our control individuals. We used only non-Finnish GnomAD individuals, as all Finnish individuals in GnomAD are included in our control exome cohort. Variants were classified as Other high impact variants if the variant was a PTV (in PTV constrained gene, pLI[22] > 0.95) or a damaging missense variant (MPC ≥2) in a gene that was not in the list of known genes (as above) and not observed in non-Finnish GnomAD individuals or in our control individuals. For homozygotes we used CADD[43] score > 20 to filter to putatively damaging variants, as MPC score is a measure of heterozygous constraint.

CADD was chosen as pathogenicity prediction method as CADD integrates multiple different prediction tools in to a single prediction score. CADD contains both conservation-based methods (e.g., GERP, Phastcons) as well as protein level scores (e.g., SIFT and Polyphen)[43].

In homozygote variant filtering we required that the variant was not seen as homozygous in non-Finnish GnomAD samples or in our internal Finnish controls.

In cases where we had parental exome data we further filtered the "likely pathogenic" variants if they were inherited from control parent or if clinical phenotype was clearly different than what has been reported in the literature (as assessed by clinical geneticist).

The algorithm for identifying pathogenic mutations was implemented in Hail[40] and executed in a Google Cloud dataproc cluster.

All CNVs passing QC criteria were classified as either (1) likely pathogenic, (2) other high impact variant, or (3) uncertain. A "likely pathogenic" classification was

assigned to deletions where the size was at least 1 Mb, and 500 kb for de novo deletions. All CNV types were additionally considered likely pathogenic (class i) when overlapping at least 75% with an established disease associated locus[44], or deleting an ID associated gene of interest (see above). CNVs were classified as "other high impact variant" (class 2) if both: (A) they were never seen in unaffected family members, population controls, or the high-quality variant set of the Database of Genomic Variants; and (B) they deleted a gene with a high probability of loss-of-function intolerance[22] (pLI > 0.95). Otherwise, a CNV was classified as a variant of uncertain significance (class 3).

**Polygenic risk scores.** As SNP weights we used summary statistics from GWA studies of schizophrenia[45], IQ[17], and educational attainment[18]. To avoid potential biases caused by non-random regional sampling of individuals in the GWA studies the summary statistics were generated after excluding all Finnish cohorts.

For polygenic scoring we used only well-imputed and genotyped common SNPs (Impute 2 info ≥0.9, allele frequency > 0.05). We pruned the SNPs to a subset of uncorrelated SNPs ($r^2 < 0.1$ within 500 kb) and used the remaining SNPs for calculating a polygenic risk score (PRS) for each individual by summing the product of beta from the summary statistics and the number of effect alleles (genotype dosage for imputed SNPs) over all SNPs. Our primary hypothesis testing used a PRS constructed from nominally significant variants ($p < 0.05$) in the original GWAS study. The genetic scores were standardized to z-scores using Finnish population controls.

For visualizing geographical differences in the PRSs within Finland, we subset the controls to those whose parents' birthplaces were within 100 km of each other. An individual's coordinates were set to the average of the parents' birthplaces' longitude and latitude. We smoothed the PRS across a map of Finland. At each map position we calculated weighted average by weighting each individual's PRS by the inverse of the squared distance between the map point and the individual's coordinate. Individuals within 50 km from the map point contributed equally to the map point, i.e., the full weight was given to those individuals independent of their exact distance from the map point.

**Association analysis.** To control for population stratification, we matched each case to its five genetically closest controls given by the first two PC's using the optmatch R package.

For replication and for studying the neurodevelopmental spectrum of candidate variants in the exome analysis, we identified neurodevelopmental (NDD) cases (ID, SCZ, and ASD) from the Finnish FINRISK population cohort as well as disease-specific collections sequenced in the UK10K study (SCZ and ASD) (Table 2). Each NDD case was genetically mapped to its five closest controls that were not matched to NFID patients.

For the dominant association analysis, we used both Fisher's exact test and Firth bias corrected logistic regression using the four first PC's as covariates. We meta-analyzed the results across the three cohorts (NFID, North NDD and South NDD) using Mantel–Haenszel meta-analysis (rma.mh in metaphor[46] R package) for Fisher's analysis and a sample size weighted meta-analysis for Firth[47].

For the recessive analysis we used a recessive allele frequency test (RAFT)[48], which takes the population allele frequency of the variant tested into account to estimate the probability of observing as many cases and controls as homozygotes under the null. As we genetically matched all cases to controls we present the analysis results from Fisher's exact test and present Mantel–Haenszel meta-analysis and Firth results in the supplement.

Association analyses were performed using Hail[40] and executed in a Google Cloud dataproc cluster.

**Enrichment analysis.** For testing if different classes of variants were enriched in cases vs. controls we used Fisher's exact test and for significant variant classes we estimated the variance explained by Nagelkerke's pseudo $r^2$.

For the CNV analysis, we used the same cases and controls as in the exome analysis where GWAS data was available (433 cases and 1100 controls passing QC for CNV analysis). Association analysis was performed testing carrier ratios using Fisher's exact test. The relevant categories were: (1) CNVs overlapping one of DECIPHER's syndromic regions (2) deletions overlapping a known developmental delay gene (Supplementary Data 1), and (3) deletions overlapping a gene with high probability of protein truncating variant intolerance (pLI > 0.95)[22].

**Heritability estimation.** We estimated the variance explained by different variant categories by fitting a logistic model and computing Nagelkerke's pseudo $r^2$ from the fitted full and null models.

Case/control status was used as a dependent variable and as an explanatory variable we used either a binary indicator for presence of variant in a given category (likely diagnostic or other high impact) or a continuous variable for PRS variance estimation. As we observed geographical differences in all evaluated PRSs we corrected for the first four PCs even after genetic matching of cases and controls to account for any residual stratification (i.e., the null model included the first four PCs). Confidence intervals for $r^2$ were estimated using adjusted bootstrap percentile method[49] by drawing 5000 bootstrap samples and computing the $r^2$ for each sample. We compared the variance explained for the whole ID cohort and also

in mild and severe ID separately. As mild and severe ID have different population prevalence we transformed the observed scale variance explained to the liability scale[50]. We used the population prevalence from a cumulative normal distribution function with mean 100 and standard deviation 15. Prevalence of 1.94%, 1.91% and 0.034% were used for all ID (IQ < 70), mild ID (50 ≤ IQ < 70) and other more severe ID combined (IQ < 50), respectively.

**Code availability**. All code used within the manuscript for all analyses is available from the corresponding author upon reasonable request.

**Reporting summary**. Further information on experimental design is available in the Nature Research Reporting Summary linked to this article.

## Data availability

All summary level data are available from the corresponding author on reasonable request. The datasets generated during and/or analysed during the current study are not publicly available due to patient confidentiality and multiple different consents of population cohorts used but subset of the data are available from the corresponding author on reasonable request. A reporting summary for this Article is available as a Supplementary Information file. The source data underlying Figs. 2, 3, 4, 5, 6b, d, f and 7 and Supplementary Figs 2-9 are provided as a Source Data file.

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

## Acknowledgements

We thank Social Science Genetic Association Consortium (SSGAC; www.thessgac.org), Psychiatric Genomics Consortium (PGC;Stephan Ripke; http://www.med.unc.edu/pgc), and Department of Complex Trait Genetics, Center for Neurogenomics and Cognitive Research (http://ctg.cncr.nl/) for sharing summary statistics of their GWAS studies. We further thank SSGAC and PGC for kindly re-generating variant weights excluding Finnish cohorts. O.P. was supported by Sigrid Juselius Foundation, Orion Research Foundation, Maud Kuistila Memorial Foundation, Brain Foundation, and Jenny ja Antti Wihuri Foundation. We also acknowledge EU/Horizon2020, COSYN, grant number 667301.

## Author contributions

M.I.K. performed the analyses; A.P., O.P., and M.J.D. conceived the study; E.S. and D.L. analyzed the CNV data; P.G. imputed the GWAS data; M.T.-H. and J.S. provided schizophrenia cohort data; A.S.H. provided Finrisk control data and identified NDD cases from population cohorts; M.M., M.Pe. and V.S. provided population control data; S.K. and M.Pi. visualized the PRS geographical distributions; J.S.M., E.R., R.K.-F., M.R., S. K.-H., J.K.-E., H.H., P.V., J.K., and O.K. collected and performed the clinical examination of the NFID cohort. E.H. collected and coordinated genetic material processing and phenotype collection. M.I.K. and A.P. wrote the manuscript. All authors critically revised the manuscript and read and approved the final version.

## Additional information

**Competing interests:** The authors declare no competing interests.

