## [Peer Review File · Nature Communications]

Reviewers' comments:

Reviewer #1 (Remarks to the Author):

Kurki and colleagues report on an analysis of variants in a Finnish intellectual disability population. The study is focused particularly on the different genetic contributions to mild versus severer forms of ID, and takes into consideration rare variants from exome sequencing, also CNVs and common variants from microarrays through polygenic risk analysis. Overall, the study is excellent, and of clear interest to the community, however, I have several concerns:

1. Given that there is a larger proportion of dysmorphic features in the severer groups (49%) compared to the mild ID (33%; Table 1), it is perhaps unsurprising that they find a higher proportion of rare damaging genetic variants in the severer ID groups.
2. The selection of a gene list based on the DDD study is somewhat surprising given the different genetic structure of the Northern Finnish population and the DDD/UK population as a whole. Would one not expect more recessive forms of disease among this population? The authors even state (line 739-740) "Consistent with the bottleneck effect and the associated Finnish Disease Heritage, we expected to observe enrichment in recessive acting variants". There are numerous studies documenting candidate ARID genes from consanguineous populations (in the literature and in OMIM), and it seems that many of these are absent from the DDD list (unsurprisingly given the population used for the DDD study), yet would perhaps be very interesting to see if variants in these genes are present in the current study. The recent paper by Hu et al (2018) includes a supplementary list compiling the findings from most of the larger studies of consanguineous ID family cohorts. This would surely be a more appropriate gene list to include in the current study.

Minor issues:

1. Line 76: Mild ID as a low extreme of a normal IQ distribution and severe ID as a distinct condition with different etiology: yes, but this is a generalization, and in family ID studies you often see a wide spectrum of severity, including mild, with the same genetic etiology.
2. Line 94: Purifying selection is common with these types of severe disorders but in North American outbred, admixed populations we still see 40% of ASD cases with ID even though it was found that many ASD genes are undergoing purifying selection
3. Line 144: Samples collected all had the same ID test performed and how was it standardized?
4. Line 198: Disability pensioners with schizophrenia were used to measure ID or as a proxy? How was ID status inferred from SCZ status?
5. Line 323: CADD was used for prioritization but were other software used in addition as ACMG guidelines suggest? why or why not?
6. Line 406: "known developmental delay gene": according to what? DDD? OMIM? Other?
7. Line 568: if the large CNV is also occurring in controls, would you not classify as variants of unknown significance?
8. Line 768: was a false discovery rate multiple testing correction attempted?
9. Line 926: There should be a statement about the limitations of exome sequencing and non coding regions that have been shown to have an effect
10. It would be interesting to know if there are differences in number/lengths of runs of homozygosity in the different regional groups.
11. Suppl Table 5 on recessive variants should not include X-linked variants.
12. It may be worthwhile pointing out that the CRADD variant reported is within the DEATH domain through which CRADD interacts with other DEATH domain proteins.
13. It should be noted that biallelic variants have also been reported previously for FRY and LAMA1 (listed in Suppl Table 5) for ARID. Maybe others too.
14. Line 895: perhaps the term "carrier" should be avoided here, as carrier is often used to describe unaffected carriers of a mutation.

15. Reference 50, Haas et al repeats the article title.

Reviewer #2 (Remarks to the Author):

With interest I read the manuscript of Kurki et al. entitled 'Contribution of Rare and Common variants to intellectual disability in a high-risk population sub-isolate of Northern Finland. The authors describe a cohort of 442 ID patients of whom the majority has mild ID. For these patients, exome sequencing and microarray analysis was performed to identify rare variants (including SNVs from exomes and CNVs from arrays) explaining disease. For patients whose diagnoses could not be established the authors used information of common variants to establish their contribution to disease etiology. Their analyses also identified mutations in CRADD as cause of a seemingly novel ID syndrome characterized by ID and pachygyria.

Major concerns

Firstly, the topic of common variation contributing to rare disease is very timely, and provides a very nice merging to two worlds: those working on rare disease and rare variants from WES and arrays, and those on common disease and common variants identified from GWAS studies. The manuscript however is overly complex as it addresses many different topics, all with relevant, but very diverse conclusions. There are three main messages: we established the diagnostic yield by rare variants in an ID cohort, we determined the contribution of rare alleles, and we identified a novel ID syndrome. In my opinion, all these messages deserve their own manuscript. Moreover, for some of the strategies described, I have some major remarks. In addition, some of the numbers mentioned throughout the manuscript do not seem to add up when comparing the numbers in figures, tables and those in the mentioned in the text, which makes cross-referencing complex.

Despite its elegant approach to determine the contribution of common variants on the same cohort which was tested 'diagnostically negative' for rare variants, the interpretation of rare variants that contribute to the diagnosing this cohort deserves more attention. The authors mention mechanistic differences between moderate/severe ID and mild ID, in which the first is merely caused by de novo mutations, and the latter not. Yet, from their approach, I get the impression that they have only sequenced singleton cases, and not trios. Given that for 239/498 (48%) of their cohort, trio-based sequencing is warranted to help detect the find those causal variants, this seems not to have been performed, yet claims on diagnostic yield is this cohort is provided. I also found it confusing to find 433 patients in the abstract and elsewhere in the manuscript, but 498 patients in Table 1. Also, whereas I understand that the population structure predisposes to homozygous mutations for recessive disease (over compound heterozygous events), the evaluation of compound heterozygous mutations is entirely omitted. Without parental allelic information from WES, it is very difficult to determine whether the variants are indeed present in bi-allelic state.

Please provide a clear tabular overview (patient-based) who has had what type of testing, and who was used in the evaluation of what numbers (also helps to make lines 161-166 more concise). Also, please indicate whether these patients were from families with negative history for ID, or not, as this information provides important clues to what type of genetic mutations one may expect as cause of disease. Lines 170-179 mentions selection of individuals with NDD from 5904 individuals for who WES was performed. How many patients were selected?

The authors also performed CNV analysis, and focus on diagnostically relevant CNVs. In this

manuscript the analysis seems solely to have focussed on deletions, yet, rare (de novo) duplications are also an important contributor to diagnose patients with ID. I can understand that perhaps throughout the manuscript, much attention is paid to PTV variants, and that CNV-losses would mimic these, but for diagnostic relevance, duplications are important too! Here again, from a diagnostic point of view, analysis of parental samples is important. In lines 232 it is mentioned that 'known artefacts' were excluded. What is the definition of 'a known artefact'. Something specific to their procedure and/or array used? Or based on the CNV overlap for complex genomic regions? The definition ≥ 9250 base pairs per SNP also raises questions, as coverage is usually indicated by 'X SNPs per xx base pairs'.

Regarding variant classification to identify the patients with a (PTV) mutation in known genes for cognitive impairment, the authors mention having used the ACMG guidelines for evaluation of variant pathogenicity. Table S2 however does not contain the evaluation of this guideline at variant level. In addition, the table does not even list the variant information at cDNA and protein level for evaluation, nor the information of which variant was identified in what patient (that is, did some patients carry two variants?). Most importantly, some of the variants listed are definitely NOT the cause of disease, simply because a PTV was identified, but the pathophysiological mechanism is not haploinsufficiency, but gain-of-function. And also, very likely, a substantial fraction of the reported missense mutations will turn out not to be the cause of disease if parental samples were tested for their inheritance pattern. So the overall report in the result section that 64 patients (but 65 in figure 2?) had a likely pathogenic variant explaining their disease is not justified. Funnily enough, this diagnostic yield of ~15% is very low when compared to previous studies consistently reporting on a diagnostic yield of ~30% for WES and an additional 10-15% for CNVs. The authors may could argue that this is due to the large fraction of mild ID cases, but in its current format, the reader cannot distinguish between these categories. That is, it is not shown which of the 64 patients had mild/moderate or severe ID.

Line 591: '85 patients (19.6%) obtained a likely diagnosis when combining WES and CNV analysis; 64 WES patients + 25 CNV patients= 89? Please explain? Is this due to multiple possible pathogenic variants in individual?'

The next approach of the authors to look at an overrepresentation of PTVs in the cohort versus controls has already been reported in many studies, both for PTVs in known disease genes as well as in for novel candidate NDD genes. Similar studies have been done at CNV level.

From lines 606, the manuscript reports on interesting new findings, that is: the contribution of common variants (in a PRS) as cause of (mild) ID. However, as the patients used in the analysis greatly depend on the exclusion of the patients with a diagnosis and given my doubts for these analyses, I cannot judge how big the impact of adjusting these deficits would be on the outcome for the PRS evaluation.

Lines 738 and onwards on the Recessive variants in Finland describe an interesting hypothesis on how the population structure of Finland would help to identify recessive disease alleles (although not novel given that inbred populations are more often used to this end). The identification of CRADD as example is very nice. However, from reading the abstract, I assumed that CRADD was a novel ID gene, and also in the results section, there is no mention that CRADD mutations are associated with disease, and NDD in particular. It is not until the discussion section that it becomes apparent that the gene is a known NDD disease gene (when mutated). Why was the CRADD gene not part of the 818 gene list evaluated (Suppl T1)? Conversely, even if present, one would have likely filtered the variant given its frequency in the Finnish population.

Minor

- Consistent use of samples/subjects/individual/participant/cases.
- Line 275: type 'included'
- Lines 625-636 seem to be redundant? Are these in the right place? Or should they have preceded line 591 or part of the Material and Methods??

Reviewers' comments:

Reviewer #1 (Remarks to the Author):

Kurki and colleagues report on an analysis of variants in a Finnish intellectual disability population. The study is focused particularly on the different genetic contributions to mild versus severer forms of ID, and takes into consideration rare variants from exome sequencing, also CNVs and common variants from microarrays through polygenic risk analysis. Overall, the study is excellent, and of clear interest to the community, however, I have several concerns:

1. Given that there is a larger proportion of dysmorphic features in the severer groups (49%) compared to the mild ID (33%; Table 1), it is perhaps unsurprising that they find a higher proportion of rare damaging genetic variants in the severer ID groups.

This is a good point raised by the reviewer! We repeated the analysis of rate of identifying different classes of variants in only those cases that are not reported to have dysmorphisms. The OR between severe vs. mild id in non-dysmorphic group was 1.88, slightly lower than when all patients are compared (OR 2.39). The difference stems mainly from the fact that in the severe non-dysmorphic ID patient group 19% of the patients had a "likely diagnostic" variant whereas in dysmorphic patients it was higher, 34%. We added this subgroup analysis as Supplementary Figure 9 and added the following paragraph to results: "As dysmorphic features are present more often in severe ID than in mild ID we wanted to analyze if "likely pathogenic" variants would be found more often in more severe ID due to dysmorphisms and not due to more severe ID. We repeated the enrichment analysis of our variant classification while restricting only to patients with no dysmorphisms (n=234) (Supplementary Figure 9). We observed "likely pathogenic" variants in 16/83 (19%) and 17/151(11%) among severe ID and mild ID patients respectively (OR: 1.88, p: 0.12). The rate of identifying "likely pathogenic" variants was lower in non-dysmorphic, severe (19%) and mild (11%) ID patients than in dysmorphic severe (34%) and mild (18%) patients. There seems to be a higher rate of "likely pathogenic" variants in severe ID patients than mild ID patients even among patients for which no dysmorphic features were recorded, although the difference is less pronounced."

2. The selection of a gene list based on the DDD study is somewhat surprising given the different genetic structure of the Northern Finnish population and the DDD/UK population as a whole. Would one not expect more recessive forms of disease among this population? The authors even state (line 739-740) "Consistent with the bottleneck effect and the associated Finnish Disease Heritage, we expected to observe enrichment in recessive acting variants". There are numerous studies documenting candidate ARID genes from consanguineous populations (in the literature and in OMIM), and it seems that many of these are absent from the DDD list (unsurprisingly given the population used for the DDD study), yet would perhaps be very interesting to see if variants in these genes are present in the current study. The recent paper by Hu et al (2018) includes a supplementary list compiling the findings from most of the larger studies of consanguineous ID family cohorts. This would surely be a more appropriate gene list to include in the current study.

The gene list from DDD study is not only genes identified in the DDD study but a list curated from literature by a team of DDD-study geneticists, including evidence from a large number of populations and inheritance patterns. Finnish population does not have inbreeding per se (e.g. cousin marriages are illegal) but due to isolated nature Finnish sub-populations are genetically closer to each other on average than in older non-isolated populations. Due to this the studies on consanguineous families are not comparable to our study. The contribution of recessive variants to ID in general is low (estimated 3.6% in 4,318 European ancestry cases in DDD study for which no de novo dominant genetic cause had not been found, <https://www.biorxiv.org/content/early/2017/11/14/201533>). We reviewed some of the genes in Supplementary Table 8 in Hu et al 2018 and on many occasions, there was only one reported case for each variant/gene providing relatively weak evidence for genes involvement in recessive developmental delays. We wanted to use a gene list with well-established link to developmental delays, which the list we used is arguably one of the best and most used such lists. We hope this clarifies the logic why we used the DDD list.

Minor issues:

1. Line 76: Mild ID as a low extreme I a normal IQ distribution and severe ID as a distinct condition with different etiology: yes, but this is a generalization, and in family ID studies you often see a wide spectrum of severity, including mild, with the same genetic etiology.

This is a good point to keep in mind. It is a generalization indeed suggested in the referenced article. Off course even Mendelian type of variants have varying degrees of penetrance in terms of phenotype spectrum. We now clarified the sentence so that it is clear for the reader that the statement refers to the Reichenberger et al paper from 2016.

2. Line 94: Purifying selection is common with these types of severe disorders but in North American outbred, admixed populations we still see 40% of ASD cases with ID even though it was found that many ASD genes are undergoing purifying selection

Many ASD patients do indeed have ID in outbred populations.

We still would argue that it is very unlikely though that individual variants that are causal in a subset of those patients are anything but extremely rare or *de novo*. If we understand the reviewer's concern correctly, the argument is that even if the enrichment of deleterious variants in bottleneck population is a well demonstrated finding, also in more outbred populations strong acting ID variants might exist. We agree, that indeed such variants do also exist in outbred populations, however, as simulated in the Zuk et al., such variants are extremely unlikely to be anything else but extremely rare in older non-bottleneck populations (Fig 3 in Zuk et al).

3. Line 144: Samples collected all had the same ID test performed and how was it standardized?

The NFID cohort cases have been evaluated and examined clinically by multiprofessional teams. Depending on the situation in question the team may consist of psychologist, physician, speech and occupational therapist, physiotherapist, nurse and social worker. Standardized IQ tests that were mainly used included different versions of following tests: Wechsler Preschool And Primary Scale Of Intelligence (WPPSI), Wechsler Intelligence Scale for Children (WISC) and Wechsler Adult Intelligence Scale (WAIS) for adults.

In case of autism spectrum disorder the diagnoses were also based on multiprofessional evaluation and different, clinically used methods such as ADOS (Autism Diagnostic Observation Schedule), ADI-R (Autism Diagnostic Interview) and CARS (Childhood Autism Rating Scale).

We added the above to the end of third to last paragraph of **Samples** chapter in the materials and methods section.

4. Line 198: Disability pensioners with schizophrenia were used to measure ID or as a proxy? How was ID status inferred from SCZ status?

Disability pension for schizophrenia was only used to see if the same regional pattern holds for schizophrenia as we observed for intellectual disability as has previously been reported (Perälä, J, 2008). We observed the same pattern as previously reported giving confidence in using disability pension data to infer relative prevalence. These data are aggregate data from the National Pension Register, we do not have access to individual level data to identify comorbidities (whether the same person would have both ID and schizophrenia). Yet, the pension is given based on one main diagnosis, so it is unlikely that there would be overlap between the two prevalence figures. Now, the topic sentence of the paragraph has been edited.

5. Line 323: CADD was used for prioritization but were other software used in addition as ACMG guidelines suggest? why or why not?

CADD is a method that integrates multiple different prediction tools in to a single prediction score. CADD contains both conservation-based methods (e.g. GERP, Phastcons) as well as protein level scores (e.g. SIFT and Polyphen). No single prediction method is superior in all settings so we chose an ensemble method to rank variants roughly to 'damaging' category and then used our own population controls and GnomAD as additional filtering. We understand the limitation of such prediction methods, but considered the CADD score as one practical strategy to filter variants.

We added the reason for choosing CADD as prediction method in to the **Identifying Likely Pathogenic Variants** chapter: "CADD was chosen as pathogenicity prediction method as CADD is a method that integrates multiple

different prediction tools into a single prediction score. CADD contains both conservation-based methods (e.g. GERP, Phastcons) as well as protein level scores (e.g. SIFT and Polyphen)²⁷.

6. Line 406: “known developmental delay gene”: according to what? DDD? OMIM? Other?

Thank you for catching this unclear reporting, we now have added the reference to Supplemental Table 1. The list of genes considered as known are shown in Supplementary Table 1 and includes genes curated from literature by the DDD study as well as genes with significant enrichment in latest developmental delay meta-analysis. See “Identifying likely pathogenic mutations” chapter.

7. Line 568: if the large CNV is also occurring in controls, would you not classify as variants of unknown significance?

Here, the CNVs identified in controls are not the same (non-overlapping) as those identified in cases. In Fig 4 we report the aggregate number of deletions, irrespective of location. So, variants in this class might not have the same location in cases and controls. We tried to explain this in “copy number variants” chapter, paragraph starting “large deletions” in the sentence: “The syndromic CNVs identified in controls were non-ID associated (12p13.33 deletion) and region with known variable phenotype (22q duplication syndrome)”. Depending on the location, even large CNVs can be tolerated without causing developmental delays.

8. Line 768; was a false discovery rate multiple testing correction attempted?

We did not use false discovery rate but we used Bonferroni correction as we wanted to be on the conservative side when reporting new findings. We added Bonferroni multiple testing correction to the sentence.

9. Line 926: There should be a statement about the limitations of exome sequencing and non coding regions that have been shown to have an effect

We have added the following statement as the next to last paragraph in the discussion where the reference is to the Nature 2018 DDD study:

“Limitations of the study includes the fact that we used exome sequencing although non-coding rare variants also contribute to the etiology of ID”⁵².

10. It would be interesting to know if there are differences in number/lengths of runs of homozygosity in the different regional groups.

We have not specifically looked at runs of homozygosity, but in the paper by Martin et al we looked at regional differences in IBD sharing, which as expected revealed large sharing in the North-Eastern parts of the Country, as expected (Fig 1 in Martin et al). The reference has been added to the Introduction.

11. Suppl Table 5 on recessive variants should not include X-linked variants.

Thank you for notifying this! One recessive variant has been removed from the supplementary table 5 and corrected the number of nominally significant variants to 57 on line 764 on the main manuscript.

12. It may be worthwhile pointing out that the CRADD variant reported is within the DEATH domain through which CRADD interacts with other DEATH domain proteins.

Thank you for the excellent suggestion. We have added the following sentence to “Recessive variants enriched in Finland chapter” in the Results section

“The variant is located in the DEATH domain through which CRADD (also known as RAIDD) interacts with other DEATH domain proteins (see ⁴¹ for a review)”

13. It should be noted that biallelic variants have also been reported previously for FRY and LAMA1 (listed in Suppl Table 5) for ARID. Maybe others too.

We have added annotations “known disease” and “known inheritance pattern” to suppl. Table 5. As the “known genes” list we used our supplementary table 1.

14. Line 895: perhaps the term “carrier” should be avoided here, as carrier is often used to describe unaffected carriers of a mutation.

We changed that sentence to “We also observed three cases that had the same missense variant in homozygous state in the INTS1 gene (Error! Reference source not found).”

15. Reference 50, Haas et al repeats the article title.

Thank you excellent catch! This was due to erroneous import to reference management software used. The duplicate title has now been removed.

Reviewer #2 (Remarks to the Author):

With interest I read the manuscript of Kurki et al. entitled ‘Contribution of Rare and Common variants to intellectual disability in a high-risk population sub-isolate of Northern Finland. The authors describe a cohort of 442 ID patients of whom the majority has mild ID. For these patients, exome sequencing and microarray analysis was performed to identify rare variants (including SNVs from exomes and CNVs from arrays) explaining disease. For patients whose diagnoses could not be established the authors used information of common variants to establish their contribution to disease etiology. Their analyses also identified mutations in CRADD as cause of a seemingly novel ID syndrome characterized by ID and pachygyria.

Major concerns

Firstly, the topic of common variation contributing to rare disease is very timely, and provides a very nice merging to two worlds: those working on rare disease and rare variants from WES and arrays, and those on common disease and common variants identified from GWAS studies. The manuscript however is overly complex as it addresses many different topics, all with relevant, but very diverse conclusions. There are three main messages: we established the diagnostic yield by rare variants in an ID cohort, we determined the contribution of rare alleles, and we identified a novel ID syndrome. In my opinion, all these messages deserve their own manuscript. Moreover, for some of the strategies described, I have some major remarks. In addition, some of the numbers mentioned throughout the manuscript do not seem to add up when comparing the numbers in figures, tables and those in the mentioned in the text, which makes cross-referencing complex.

We thank the reviewer for appreciating our approach. We agree that the manuscript has more ‘angles’ than typical manuscripts in the field. However, our aim was to comprehensively characterize our unique cohort. We admit that this makes it harder to grasp all messages in one reading, but argue that there should be more such comprehensive descriptions of developmental delay cohorts, and not just reports that “pick candies from the cake”. We find this especially relevant in this case, where mild ID represents a big fraction of the cohort and the genetic background of mild ID is less well established. We’ll address the reported discrepancies below.

Despite its elegant approach to determine the contribution of common variants on the same cohort which was tested ‘diagnostically negative’ for rare variants, the interpretation of rare variants that contribute to the diagnosing this cohort deserves more attention. The authors mention mechanistic differences between moderate/severe ID and mild ID, in which the first is merely caused by de novo mutations, and the latter not. Yet, from their approach, I get the impression that they have only sequenced singleton cases, and not trios.

This is an important point in definitive diagnosis of ID patients with no family history. We had in fact trios from 138/442 exome sequenced patients (For clarification 433 is the number with both CNV and exome data, which is reported in “total diagnosis rate” chapter). Due to the relatively low number of full trios available (about 1/3 of cases) we initially decided to follow the same diagnostic criterion for the entire cohort in order not to bias our analyses. That is why we called the highest diagnostic category “likely pathogenic variant identified” to reflect the uncertainty. We used a quite large reference sample from the same population (jointly called with the NFID patient) to filter out most inherited variants (although close to family specific variants could slip through). The uncertainty in variant identification is exactly the same in mild and more severe categories and therefore does not affect the conclusion that damaging variants in known genes explain more of the severe patients than mild ID patients.

Regardless, we have now used and report trios and duos for which we have exome sequence data to refine the list of “likely diagnostic” patients. We have added indicators if the patient was part of a trio or a duo and if the variant was confirmed de novo or confirmed not inherited from the other parent. 4/64 patients had inherited variants (4/23 in trios). This information is now added in Supplementary Table 2 and new Supplementary Table 3. We had 26 patients with “likely diagnostic” variant but without inheritance information. As shown by doing the filtering first and then checking the inheritance pattern, the parental info does not change the interpretation in the majority of the cases (~83%). We added information on availability of parental information and updated diagnostic column “likely diagnostic” after the parental information has been taken in to account on Supplementary Table 2 and added comment column.

We updated all the figures and numbers to reflect the updated “likely diagnostic” classification that takes available parental information into account.

Changes reflecting the above:

- 2nd paragraph of results section we added: “Out of the 422 independent patients we had exome data for 138 full trios, 133 duos and the remaining 171 patients were cases only.”
- End of first paragraph in **Identifying likely pathogenic mutations** method chapter we added: “In cases where we had parental exome data we further filtered the “likely pathogenic” variants if they were inherited from control parent or if clinical phenotype was clearly different than what has been reported in the literature (as assessed by clinical geneticist).” First paragraph of Mutations in known genes causing cognitive impairment chapter in results section: “For the subset of individuals for which we had parental exome available (138 trios and 133 duos) we further filtered the list of “likely pathogenic” variants by not being inherited from a parent without learning disability. This step filtered 4/24 likely pathogenic variants in trios and 0/15 in duos (Supplementary Table 2). We also excluded 2 “likely diagnostic” variants because the clinical phenotype was clearly different (as assessed by clinical geneticist) than what has been reported in the literature. After these filtering we identified “likely pathogenic” diagnosis for 60/422 patients in exome sequencing.”
- Updated Figure 2 to reflect the new classification above.
- Updated ORs and p-values at the end of **Mutations in known genes causing cognitive impairment** chapter to reflect the updated classification
- Updated ORs and p-values in the beginning of **Total genetic diagnosis rate chapter**
- Updated supplementary Figure 7 referenced in the beginning of **Total genetic diagnosis rate chapter**
- Updated supplementary Figure 8
- Updated the percent of “likely pathogenic” patients in the beginning of **Polygenic common variant burden** chapter
- Updated the percent of “likely pathogenic” patients in the beginning of Variance explained by different variant categories chapter
- Added to the limitations of the study at the end of second to last paragraph of **discussion**: “Also, we did not have exome sequencing for parents for 2/3 of the patients, but we performed sensitivity analyzes on the subset of patients, where we had full trios. These supported the conclusions that common variant polygenic load and rare variants might act additively and that mild ID is less affected by de novo/extremely rare variants in known ID genes (See Total genetic diagnosis rate chapter and Supplementary Figure 4).”

In order to assess the impact of misclassifying “likely pathogenic” variants on the conclusion that severe ID is more affected by these than mild ID we separated patients to: 1) full trios with confirmed de novo in dominant acting gene 2) trios where we checked that the other parent did not have variants and 3) all “likely pathogenic” variants. The OR in SEVERE vs. mild ID patients having “likely pathogenic” variant was 2.1, 2.2 and 2.3 in confirmed de novos (trios), duos and all patients respectively. This suggests that there would not be a big impact from misclassification to the conclusion that severe ID patients are more often affected by de novos/very rare variants than mild ID.

We added the following sentence to the second to last paragraph of the Total Diagnostic Rate chapter: “As we did not have parental exome sequencing data on all patients, we wanted to assess if the uncertainty in “likely diagnostic” classification affects the result that mild ID would be less affected by de novo/ultra-rare variants in known ID genes. To this end we subset the cases to 1) full trios with confirmed de novo in dominant acting gene 2) duos where we checked that the other parent did not have the variant and 3) all

patients with “likely pathogenic” variants. The OR in SEVERE vs. mild ID patients having “likely pathogenic” variant was OR 2.2 (0.6 – 9.5), 2.8 (0.9 - 8.8) and 2.2 (1.3 - 3.8) in confirmed de novos (trios), duos and all patients respectively. This suggests that some misclassification would not have a big impact on the conclusion that severe ID patients are more often affected by de novos/very rare variants in known ID genes than mild ID.”

Given that for 239/498 (48%) of their cohort, trio-based sequencing is warranted to help detect the find those causal variants, this seems not to have been performed, yet claims on diagnostic yield is this cohort is provided. I also found it confusing to find 433 patients in the abstract and elsewhere in the manuscript, but 498 patients in Table 1.

We indeed had 498 sequenced patients (including siblings and family members) but in order to report sound statistical analyses, we report the analysis of only independent cases (442 with exomes sequencing data and 433 patients with both exome sequencing and CNV), in other words, family members were excluded in this analysis. This has now been clarified by adding a sentence to Table 2: “The number of individuals in the analyses are after QC and after related individuals have been removed”

Also, whereas I understand that the population structure predisposes to homozygous mutations for recessive disease (over compound heterozygous events), the evaluation of compound heterozygous mutations in entirety omitted. Without parental allelic information from WES, it is very difficult to determine whether the variants are indeed present in bi-allelic state.

The reviewer is right. The major strength of Finnish population in these types of studies is the nature of the Finnish genetic landscape shaped by a small founder population, bottlenecks and relatively short time since the bottlenecks. Due to the unique population history some variants with negative selection can still exist at high levels compared to older populations because negative selection has not had enough time to drive the allele frequency low/out of the population. Due to the bottleneck Finnish population has less variants overall and hence the homozygous mutations are expected to be more common genetic cause than compound heterozygotes. This has been demonstrated e.g. in the case of Finnish Disease Heritage mutations, where one major mutation is mainly causing the disease (<https://www.ncbi.nlm.nih.gov/pubmed/12627297>). The contribution of recessive variants to ID in general is low (estimated 3.6% in 4,318 European ancestry cases in DDD study for which no de novo dominant genetic cause had not been found, <https://www.biorxiv.org/content/early/2017/11/14/201533>). As we did not have parent’s exomes for 2/3 and could not accurately confirm compound heterozygous state we decided not to include compound heterozygote analyses. Due to these facts any decline in diagnostic rate caused by omission of compound heterozygotes is likely to be negligible.

Please provide a clear tabular overview (patient-based) who has had what type of testing, and who was used in the evaluation of what numbers (also helps to make lines 161-166 more concise). Also, please indicate whether these patients were from families with negative history for ID, or not, as this is information provides important clues to what type of genetic mutations one may expect as cause of disease.

We added the requested information as Supplementary Table 3

Lines 170-179 mentions selection of individuals with NDD from 5904 individuals for who WES was performed. How many patients were selected?

Thank you for pointing this out as it was unclearly written. We identified 636 NDD patients, which are given in table 2 (NFNDD and SFNDD cases). We have added the N and referenced that table after the sentence. “N=636, NFNDD and SFNDD cases in Table 2”

The authors also performed CNV analysis, and focus on diagnostically relevant CNVs. In this manuscript the analysis seems solely to have focused on deletions, yet, rare (de novo) duplications are also an important contributor to diagnose patients with ID. I can understand that perhaps throughout the manuscript, much attention is paid to PTV variants, and that CNV-losses would mimic these, but for diagnostic relevance, duplications are important too! Here again, from a diagnostic point of view, analysis of parental samples is important. In lines 232 it is mentioned that ‘known artefacts’ were excluded. What is the definition of ‘a known artefact’. Something specific to their procedure and/or array used? Or based on the CNV overlap for complex genomic regions? The definition ≥ 9250 base pairs per SNP also raises questions, as coverage is

usually indicated by 'X SNPs per xx base pairs'.

We thank the reviewer for identifying this, which reflects an unclear reporting of our approach. We did include duplication analyses that were located in known pathogenic duplication regions. This has now been expressed more clearly in Materials and Methods, under statistical analyses. There was one large duplication in the Prader-Willi/Angelman region in one case (Supplementary Table 3). We have now also added the CNV type in Supplementary Table 3, we apologize for not including this in the original submission. Figure 4 panel C has been added to show duplication carrier frequencies in cases vs. controls.

As for known artefacts, these genomic regions known for their high rate of false CNV calls were reported by Jacquemont et al (Jacquemont S, 2014) in the American Journal of Human Genetics 2014. This reference has now been added in the main text in Materials and Methods under "GWAS genotyping and CNV calling"

The definition of $\geq 9\ 250$ base pairs / SNP can be restated as 1 SNP per 9 250 base pairs, or 1.08 SNPs per 10 kb. This threshold was determined for the use of the control population, based on a high likelihood of CNV calls with fewer SNPs / 10 kb than 1.08 being found false in subsequent visual confirmation of the variant calls.

Regarding variant classification to identify the patients with a (PTV) mutation in known genes for cognitive impairment, the authors mention having used the ACMG guidelines for evaluation of variant pathogenicity. Table S2 however does not contain the evaluation of this guideline at variant level. In addition, the table does not even list the variant information at cDNA and protein level for evaluation, nor the information of which variant was identified in what patient (that is, did some patients carry two variants?).

Thank you for noticing this. This was a careless omission on our part. We added the cDNA and protein level HGVS annotations to Table S2

We regret for an unclear formulation in the original manuscript. We stress that we did not use the ACMG guidelines for variant classification as the aim of this study was not a clinical grade diagnostic evaluation. In the CNV section we mention that "likely pathogenic" CNVs were "largely similar" to the ACMG classification, but not strictly so. We now have deleted the reference to ACMG guidelines also from the CNV section, to avoid confusion.

Most importantly, some of the variants listed are definitely NOT the cause of disease, simply because a PTV was identified, but the pathophysiological mechanism is not haploinsufficiency, but gain-of-function.

We went through the list in Supplementary table 2 again but could not spot a variant where the mechanism in the literature is a gain of function and we had PTV. 16:30731534:C:T PTV in SRCAP has dominant negative annotation in the used DDD annotation but looking at the reported pathogenic variants, majority of them are PTVs: (<https://decipher.sanger.ac.uk/gene/SRCAP#variants/SRCAP/patient-overlap/snvs>).

And also, very likely, a substantial fraction of the reported missense mutations will turn out not to be the cause of disease if parental samples were tested for their inheritance pattern. So the overall report in the result section that 64 patients (but 65 in figure 2?) had a likely pathogenic variant explaining their disease is not justified.

Our main aim in this manuscript was not to have 100% accurate clinical grade pathogenicity assessment for each individual patient but to strictly define damaging variants that do not exist (in dominant acting variants) or are low in allele frequency (recessive) in global control populations (GnomAD) and population specific controls jointly called with case samples (our controls). These dominant variants are the only variants that could be the cause in dominant acting cases. Using a trio approach for all could eliminate some of those in which case the patient would have been unsolved. This is because our filtering would be anyway necessary one anyway for a variant to be dominantly pathogenic. For most patients we ended up with only one possible variant.

We estimated the impact of the uncertainty of diagnosis in patients where we did not have parental exome sequencing data above. ~ 17% of the diagnoses changed after considering evidence from parent's exome.

Funnily enough, this diagnostic yield of ~15% is very low when compared to previous studies consistently reporting on a diagnostic yield of ~30% for WES and an additional 10-15% for CNVs. The authors may argue that this is due to the large fraction of mild ID cases, but in its current format, the reader cannot distinguish between these categories. That is, it is not shown which of the 64 patients had mild/moderate or severe ID.

The information in identifying likely pathogenic variant in mild vs. severe ID is given in Figure 5 C as well as for each identified likely pathogenic variant in Table S2 (column CLIN_PHENO_Level_of_ID). From Figure 5 C one can see that we identified likely pathogenic variant in 27% of severe cases whereas we identified likely pathogenic variant only in 13% of mild ID cases. As majority of our cases are mild the overall “diagnostic yield” is lower than reported in the literature but on par when considering only severe cases. This justifies the claim that the overall rate is lower due to large fraction of mild ID.

Line 591: '85 patients (19.6%) obtained a likely diagnosis when combining WES and CNV analysis; 64 WES patients + 25 CNV patients= 89? Please explain? Is this due to multiple possible pathogenic variants in individual?

This was an omission caused by removing patient IDs before submitting. Indeed 4 individuals had variants from exactly 2 categories. We have now added individual id to Supplementary Table 2.

The next approach of the authors to look at an overrepresentation of PTVs in the cohort versus controls has already been reported in many studies, both for PTVs in known disease genes as well as in for novel candidate NDD genes. Similar studies have been done at CNV level.

From lines 606, the manuscript reports on interesting new findings, that is: the contribution of common variants (in a PRS) as cause of (mild) ID. However, as the patients used in the analysis greatly depend on the exclusion of the patients with a diagnosis and given my doubts for these analyses, I cannot judge how big the impact of adjusting these deficits would be on the outcome for the PRS evaluation.

We first report the association of PRS to all forms of ID without any exclusion of patients based on diagnosis. This clearly shows that common variant polygenic load is a contributing factor to ID etiology. In sub-analysis we then asked the question if the PRS does have an effect in those patients for which a likely pathogenic variant has been identified. In our analysis it seems that those diagnosed patients are as highly affected as the non-diagnosed patients. This part is indeed susceptible to possible misdiagnosis (false negatives and false positives). This does not however affect the main point of the manuscript: the common variant polygenic load is a contributing factor in genetic etiology of ID.

In order to evaluate if our sub-analysis is biased because of uncertainty in diagnosis, we compared the PRSs between those for which a *de novo* variant (in subset of 138/442 patients for which we have full trios) in known dominant acting gene was identified. 13 patients had a *de novo* in known gene or a diagnostic CNV. We modified supplementary Figure 4 to include scores for those 13 patients. These patients are just as affected (point estimate even more so) by educational attainment and IQ PRSs as those for which no such variant has been identified. However, these patients seem not to be affected by the SCZ PRS.

We added the following statement to the end of “Polygenic common variant load”: “Importantly we also subset “likely diagnosed” patients to those for which we had confirmed *de novo* mutation in a known ID gene or diagnostic CNV (n=13). Those 13 patients were just as affected by IQ and EDU PRSs but not by SCZ PRS (Supplementary Figure 4)”.

Lines 738 and onwards on the Recessive variants in Finland describe an interesting hypothesis on how the population structure of Finland would help to identify recessive disease alleles (although not novel given that inbred populations are more often used to this end). The identification of CRADD as example is very nice. However, from reading the abstract, I assumed that CRADD was a novel ID gene, and also in the results section, there is no mention that CRADD mutations are associated with disease, and NDD in particular. It is not until the discussion section that it becomes apparent that the gene is a known NDD disease gene (when mutated). Why was the CRADD gene not part of the 818 gene list evaluated (Suppl T1)? Conversely, even if present, one would have likely filtered the variant given its frequency in the Finnish population.

Thank you for pointing out this potentially misleading way of reporting.

We used the gene list curated by DDD study as the list of known genes. CRADD was not at the time of

analysis considered as confirmed or probable ID gene in the DDD database. After observing the association, we turned in to literature and found the previous article linking *CRADD* to ID and then cited that in the discussion. The manuscript structure and flow reflect how the analysis was conducted. Indeed, it would have been filtered anyway as the frequency is markedly high in Finnish population (exactly the strength of the Northern Finnish ID cohort). We have edited the abstract to clarify that *CRADD* is a known ID gene.

Minor

- Consistent use of samples/subjects/individual/participant/cases.

- Line 275: type 'included'

Thank you for noticing the typo has now been fixed.

- Lines 625-636 seem to be redundant? Are these in the right place? Or should they have preceded line 591 or part of the Material and Methods??

Thank you for noticing this. It is indeed redundant and has now been removed as it is described in materials and methods

Reviewers' comments:

Reviewer #1 (Remarks to the Author):

Overall, the authors have done an impressive and excellent job in addressing the issues raised in the review.

My only remaining criticism is with the limitations in the DDD-based gene list used (Reviewer #1, point 2). Reviewer #2 made a related point concerning CRADD, asking why it was not included in the original list, since it is an established and corroborated ARID gene. I count at least 3 independent reports of CRADD and ARID in the literature. Adding CRADD post hoc to the analysis is fine, but there are still many other ARID genes with way more supporting evidence than CRADD that are not in the DDD list. I agree many of the genes reported in the Hu et al table have limited support, but there are still a number that seem to recur. The arguments made about Finland not having inbreeding per se despite the isolated nature of the population as a reason for not being more inclusive with ARID genes in the gene list seem to be somewhat contradicted by their report of the homozygous mutation in CRADD.

Reviewer #2 (Remarks to the Author):

Firstly, I wish to thank the authors for their revised version of their manuscript. In my opinion, they have done an excellent job in addressing the points raised by the reviewers.

I have a few remaining further questions/remarks, and some small typos to consider:

- Line 355: Class i  Class 1
- Line 461: 422 independent  442 independent
- Lines 484 - 485: seem to overlap with lines 481 and 482?
- Lines 603-604: sentence reads slightly odd? Perhaps add 'A' at the beginning of the sentence?
- In figure 4 the term 'carrier' seems inappropriately used, as this is solely used to describe an individual who has inherited a genetic (recessive) trait or mutation, but displays no symptoms. Please consider to revise.
- Lines 655 - 658: I understand the rationale that more severe ID is associated with larger chromosomal abnormalities, and that these were not included in their study as these were likely previously diagnosed. Yet the total contribution of large(r) chromosomal abnormalities to the total contribution of diagnostic yield for ID is far less than that of point mutations. Hence I do not expect this to be the (only) explanation.
- The relation between Figure 5, the diagnostic yield and suppl Table 3 needs clarification. Initially I remarked that the diagnostic yield is low when compared to other studies. The author provide an explanation that this is due to the 'comparable to other studies high yield for severe ID' and 'lower yield' for mild ID, as can be seen from Figure 5C. I understand their explanation, but do not understand how the numbers in Figure 5C and the levels of ID match with those in Suppl. T3. Must be definitions... It seems that for the category Severe 'Moderate and Profound' ID were added together to come to a total of n=180 individuals. Based on ICD10, one should distinguish Mild, moderate, severe and profound ID?
- Related to this, the authors have provided data for which patients with a likely pathogenic variant duo's or trio's were sequenced (Suppl T2). It would however be good to have that information presented for Suppl T3. Were parental WES samples available equally for Mild and Severe ID? That would allow to answer the question whether or not de novo mutations as explanation for disease were potentially missed due to the absence of parental samples for the Mild ID group, whereas these were present for the severe group?

Reviewers' comments:

Reviewer #1 (Remarks to the Author):

Overall, the authors have done an impressive and excellent job in addressing the issues raised in the review.

My only remaining criticism is with the limitations in the DDD-based gene list used (Reviewer #1, point 2). Reviewer #2 made a related point concerning CRADD, asking why it was not included in the original list, since it is an established and corroborated ARID gene. I count at least 3 independent reports of CRADD and ARID in the literature. Adding CRADD post hoc to the analysis is fine, but there are still many other ARID genes with way more supporting evidence than CRADD that are not in the DDD list. I agree many of the genes reported in the Hu et al table have limited support, but there are still a number that seem to recur. The arguments made about Finland not having inbreeding per se despite the isolated nature of the population as a reason for not being more inclusive with ARID genes in the gene list seem to be somewhat contradicted by their report of the homozygous mutation in CRADD.

It is possible that we might add few diagnoses with hand-selecting some additional recurring recessive ID genes from the literature.

However, we find that the DDD study list is one of the most curated, complete and widely used lists available. To provide consistency with the most recent literature we decided to use the DDD list as a basis.

Also, as shown in a DDD study (<https://www.biorxiv.org/content/early/2017/11/14/201533>) the contribution of recessive variants is very low (3.6%). Also, we report very few recessive variants (1.8% of cases) with the current gene list which contains 619 bi-allelic genes. Any small additions to this list would be so minor that it will not change any conclusions made in the study. However, we have now added CRADD in the list of known genes as suggested by this reviewer, Supplementary Table 1. CRADD is indicated with an asterisk to highlight that this has been added post-hoc. We hope that this is a satisfactory compromise between the two views.

Reviewer #2 (Remarks to the Author):

Firstly, I wish to thank the authors for their revised version of their manuscript. In my opinion, they have done an excellent job in addressing the points raised by the reviewers.

I have a few remaining further questions/remarks, and some small typos to consider:

- Line 355: Class i  Class 1

I changed to 1

- Line 461: 422 independent  442 independent

Thank you for catching this. Changed to 442

- Lines 484 - 485: seem to overlap with lines 481 and 482?

This was unnecessary repetition and the first sentence referring to sample numbers has been removed.

- Lines 603-604: sentence reads slightly odd? Perhaps add 'A' at the beginning of the sentence?

Agreed. 'A' has been added to the beginning of sentence

- In figure 4 the term 'carrier' seems inappropriately used, as this is solely used to describe an individual who has inherited a genetic (recessive) trait or mutation, but displays no symptoms. Please consider to revise.

Agreed, the term 'carrier' has been removed from figure 4 for clarity. We also modified figure 4 caption to remove carrier word.

- Lines 655 - 658: I understand the rationale that more severe ID is associated with larger chromosomal abnormalities, and that these were not included in their study as these were likely previously diagnosed. Yet the total contribution of large(r) chromosomal abnormalities to the total contribution of diagnostic yield for ID is far less than that of point mutations. Hence I do not expect this to be the (only) explanation.

We agree that it might not be the only explanation. In Figure 5 B we show that diagnostic CNV was identified in 6.7% of the patients whereas the sum of point mutations is 13.8%. In recent CNV analysis following clinical diagnostic the diagnosis rate was 11% for recurring CNVs (<https://www.nature.com/articles/eihg2016107>, <https://onlinelibrary.wiley.com/doi/abs/10.1111/cge.13009>) which is not a lot less than what we report. It is clear that

we have a smaller fraction of CNV diagnoses, at least partly because of pre-screening that caused an ascertainment bias. The CNV diagnosis rate is similar between mild and more severe in our data but whether it is just because of more severe ID has been excluded based on previously detected chromosomal abnormalities cannot be conclusively determined.

We modified the sentence and added one sentence to read:

“One cause of this is that a large fraction of ID patients who had a chromosomal abnormality had been identified in previous clinical cytogenetic analyses and thus excluded from this study. Whether the CNV diagnostic rate is truly similar in mild and more severe ID cannot be conclusively determined from our data.”

- The relation between Figure 5, the diagnostic yield and suppl Table 3 needs clarification. Initially I remarked that the diagnostic yield is low when compared to other studies. The author provide an explanation that this is due to the 'comparable to other studies high yield for severe ID' and 'lower yield' for mild ID, as can be seen from Figure 5C. I understand their explanation, but do not understand how the numbers in Figure 5C and the levels of ID match with those in Suppl. T3. Must be definitions... It seems that for the category Severe 'Moderate and Profound' ID were added together to come to a total of n=180 individuals. Based on ICD10, one should distinguish Mild, moderate, severe and profound ID?

It is a relevant point that the ICD classification has more granularity than “moderate and profound”. However, in analysis our aim was to highlight the difference between the mild and the other more severe ID forms. To this end we combined other categories as one ‘severe’ category in Figure 5. Using only two groups makes also the groups larger and thus provide more statistical power to highlight the difference seen in mild ID. In Supplementary Table 3 we mistakenly used less granular classification and severe and profound were all in profound category. This has now been fixed in Supplementary Table 3

- Related to this, the authors have provided data for which patients with a likely pathogenic variant duo's or trio's were sequenced (Suppl T2). It would however be good to have that information presented for Suppl T3. Were parental WES samples available equally for Mild and Severe ID? That would allow to answer the question whether or not de novo mutations as explanation for disease were potentially missed due to the absence of parental samples for the Mild ID group, whereas these were present for the severe group?

The information if patient was part of trio or duo has now been added to Suppl T3. We had trios for 74 (32.2%) of MILD ID patients and 53 (28.8%) of more SEVERE patients. We had duos for 79 (34.4%) mild patients and 45 (25.5%) for more SEVERE ID. Thus we had similar proportions of familial information for both ID severities groups.

We have added the following statement at the end of **Total genetic diagnosis rate chapter**:

“We further had similar proportions of parental data available for mild and more SEVERE ID patients. We had 74 (32.2%) trios and 79 (34.4%) duos in the mild ID patient group and 53 (28.8%) trios and 45 (25.5%) duos in more SEVERE ID patients.”

REVIEWERS' COMMENTS:

Reviewer #1 (Remarks to the Author):

I am satisfied with the authors' response.
No further comments.

Reviewer #2 (Remarks to the Author):

I wish to thank the authors once more their answers on the comments I made. At this points I have no further suggestions or remarks.